# Seasonal evolution of snow density and its impact on thermal regime of sea ice during the MOSAiC expedition

Yubing Cheng<sup>1</sup>, Bin Cheng<sup>2</sup>, Roberta Pirazzini<sup>2</sup>, Amy R. Macfarlane<sup>3,4</sup>, Timo Vihma<sup>2</sup>, Wolfgang Dorn<sup>1</sup>, Ruzica Dadic<sup>5</sup>, Martin Schneebeli<sup>5</sup>, Stefanie Arndt<sup>6</sup>, and Annette Rinke<sup>1</sup>

5 Alfred Wegener Institute, Helmholtz Centre for Polar and Marine Research, Potsdam, Germany

<sup>2</sup>Finnish Meteorological Institute, Helsinki, Finland

<sup>3</sup>The Arctic University of Norway UiT, Tromsø, Norway

<sup>4</sup>Northumbria University, Newcastle Upon Tyne, UK

<sup>5</sup>WSL Institute for Snow and Avalanche Research SLF, Davos Dorf, Switzerland

<sup>6</sup>Alfred Wegener Institute, Helmholtz Centre for Polar and Marine Research, Bremerhaven, Germany

Correspondence to: Bin Cheng (bin.cheng@fmi.fi)

Abstract. Snow density is a crucial parameter for sea ice modelling at the physical process level. The seasonal evolution of surface (top 3 cm) and bulk (entire layer) snow densities observed during the MOSAiC expedition was investigated and used to assess four snow density schemes. A numerical snow and sea ice model was applied to simulate the sensitivity of sea ice to snow density and snow precipitation during the period when snow was dry. Snow densities of 348 kg/m³, 308 kg/m³, and 487 kg/m³ were derived from linear regression of snow water equivalent (SWE) against snow depth, using samples collected during three distinct periods: the entire MOSAiC period, the winter-spring period (October-May), and the summer-autumn period (June-September), respectively. The examined snow density schemes produced mean snow densities consistent with MOSAiC observations; however, none of the schemes adequately captured the observed temporal variability in snow density. The modelled mean snow surface temperature and ice thickness showed a linear relationship with snow density, whereas the modelled mean in-snow and in-ice temperatures showed an inverse linear relationship with snow density. The impacts of a time-dependent snow density on snow and ice thermodynamic regimes were stronger than in the model runs using a constant snow density during the model period. Model sensitivity experiments showed that a higher snow density reduces snow and ice temperatures, promoting ice growth, whereas increased snow precipitation has the opposite effect. However, excessive snow accumulation can thicken the ice due to snow-ice formation.

#### 1 Introduction

Snow on top of sea ice is an important component of the marine Arctic climate system. Snow is a strong reflector of solar radiation, with a surface albedo much higher than that for bare ice (Perovich and Polashenski, 2012). This keeps the surface

temperatures lower in spring and autumn and reduces the melt in late spring and early summer. Snow is a good insulator, having a heat conductivity of only approximately 10-20 % of that of sea ice (Sturm et al., 2002a; Macfarlane et al., 2023a). Hence, in winter, the heat flux from the relatively warm ocean to the cold atmosphere is much smaller when sea ice is covered by snow, resulting in a reduction in the ice growth rate (Merkouriadi et al., 2017). On the contrary, snow has a positive contribution to ice thickness via snow-to-ice transformation. This occurs when the ice surface is flooded under a heavy snowpack and slush freezes to form snow ice (Provost et al., 2017) or when meltwater or rain percolates to the snow-ice interface and refreezes there to form superimposed ice (Cheng et al., 2003). A surface scattering layer (SSL) forms in summer and persists beyond it, buried below new accumulated snow. SSL visually looks like a snow layer. The isotopic signature of SSL is purely from sea ice. The meltwater drains to melt ponds and leads, leaving the SSL relatively dry. As the top SSL melts, there is a simultaneous transformation of the underlying bare ice to the SSL. Thus, the thickness of the SSL remains almost stable throughout the summer, and its density increases with depth (Macfarlane et al., 2023b). SSL reveals a porous structure that resembles snow (Smith et al., 2022). In winter, the densities of snow and the SSL affect the thermal properties of the snowpack (Macfarlane et al., 2023b).

Previous studies have shown that snow density and its potential to be eroded by wind are correlated: the lower the snow density, the weaker the wind required for erosion and drifting (He and Ohara, 2017). Drifting snow can enhance sublimation (Sigmund et al., 2022) and act as a source of fine-mode salt aerosols, which serve as cloud condensation nuclei (Gong et al., 2023). In satellite retrievals of sea ice thickness, snow density is critically important because vertical gradients in snow density and/or volume scattering in the snow influence the radar signal (Kern et al., 2015).

40

After snowfall, snow density begins to evolve due to mechanical compaction and snow metamorphism, generally leading to densification of snow (Bormann et al., 2013; Helfricht et al., 2018; Judson and Doesken, 2000). Snow metamorphism is controlled by dynamic and thermodynamic factors. The dynamic factors are wind (decomposition, erosion and redeposition) and snowfall (weight). Wind-driven processes result in the breaking of the snow structure and generally leading to increased snow density (King et al., 2020). New snow usually has lower density than old snow (Sturm and Holmgren, 1988), reducing the bulk density of the snowpack. The major challenge in quantifying the role of snowfall in snow metamorphism, snow depth, and ice mass balance beneath the snow layer is that the temporal and spatial variations of snowfall are often poorly understood. Measurements of the total local seasonal snowfall tend to differ from each other, depending on the instrument used (Matrosov et al., 2022). The rearrangement of snow microstructure during snow metamorphism affects snowpack density. Sublimation and deposition of water vapour under a temperature gradient can drive kinetic growth of faceted crystals, which may inhibit compaction, whereas other metamorphic processes can enhance bonding and increase snow density (Jafari et al., 2020; Nicolaus et al., 2022).

Significant advancement has been made in the representation of the microscale physics controlling snow density (Keenan et al., 2021); however, major challenges remain in representing snow processes in climate models. Those can only resolve bulk

properties of the snowpack, such as its depth and temperature. Various snow-density parameterisations have been developed (e.g., Essery et al., 2013; Keenan et al., 2021). In most climate models involved in the Snow Model Intercomparison Project (SNOW-MIP), snow density is either constant or depends on mechanical compaction and snow age (Menard et al., 2021). However, limited and sparse data availability from the central Arctic has hindered a comprehensive assessment of the snow density on sea ice and its dependence on variables resolved by climate models.

In this study, we investigate the seasonal evolution of snow density during the Multidisciplinary Drifting Observatory for the Study of Arctic Climate (MOSAiC) expedition. We address snow density for the surface snow layer (defined by the sampling resolution of a density cutter, here the top 3 cm) and the bulk snowpack (entire snow layer). The surface snow layer is characterised by new snow deposition. It is highly sensitive to atmospheric influences, responds more rapidly to variations in the surface energy balance than deeper layers, and influences the surface energy balance through its radiative properties and surface roughness. The bulk snowpack density is critical for sea ice mass balance. Our objectives are to find out (1) how snow density is affected by air temperature during the MOSAiC expedition, (2) how snow density is affected by wind during the winter-spring period, (3) how the simulated thermal evolution of snow and ice is affected by application of different snow density schemes, and (4) how snowfall impacts the thermal evolution of the previously deposited snow and underlying sea ice.

To fulfil the objectives, we analyse MOSAiC snow density and weather observations, evaluate snow density schemes, and model snow and sea ice thermodynamics. In modelling, we focus on the winter-spring (October – May) period, when the MOSAiC ice camp drifted along a continuous trajectory (MOSAiC legs 1-3).

## 2 Data and methods

The MOSAiC expedition to the central Arctic started in October 2019 and lasted until September 2020. The MOSAiC ice camp (RV *Polarstern*) drift started from the marginal ice zone (MIZ) in the eastern Amundsen Basin and ended in the Fram Strait (Fig. 1). The expedition was divided into five legs. Legs 1 to 3 were operated along a continuous ice camp (central observatory 1, CO1) between late October 2019 and mid-May 2020. Leg 4 took place between June and August 2020 (Central Observatory 2, CO2). Leg 5 was set up after a complete relocation of RV *Polarstern* back up to the central high-Arctic (central observatory 3, CO3). The entire MOSAiC drift trajectory is given in Fig. 1.

Figure 1: (a) The MOSAiC ice camp drift trajectory during legs 1, 2, and 3. (b) Evolution of observed mean bulk snow density with respect to time and latitude. The colour bar illustrates the density values in both (a) and (b). Note that observations were also taken out of the ice camp trajectory at temporary stations (individual dots in both plots) established after the MOSAiC expedition ended on 17 September 2020.

# 2.1 Weather data





Meteorological variables were measured at the Central Observatory Met City at approximately 500 m distance from RV Polarstern (Shupe et al., 2022), where a 10-m-high weather mast was installed. We use 2-m air temperature and 10-m wind speed to investigate their impact on snow density since those are fundamental meteorological variables affecting snow metamorphism (Sommerfeld and LaChapelle, 1970; Domine et al., 2007). Precipitation measurements were taken using different sensors located onboard the RV Polarstern and at Met City site on the sea ice (Shupe et al., 2022). The snow precipitation measurements suffered from different artefacts, some of the onboard measurements being more affected by the ship superstructure, while on-ice observations were more influenced by blowing snow. Intercomparison studies concluded that the most reliable and consistent observations were obtained from the vertically pointing 35-GHz Doppler radar and the optical disdrometer onboard the RV Polarstern (Matrosov et al., 2022; Wagner et al., 2022). However, in this study, we utilised the monthly cumulative precipitation measurements presented by Matrosov et al. (2022) to assess the sensitivity of modelled snow and ice parameters to precipitation uncertainty. For modelling, we used the hourly ERA5 reanalysis products (Hersbach et al., 2020) of wind speed  $(V_a)$ , air temperature  $(T_a)$ , relative humidity (Rh), precipitation rate (P), as well as shortwave  $(Q_s)$  and longwave  $(Q_l)$  radiative fluxes, extracted from the grid cells closest to the drift track coordinates of the RV *Polarstern*. The comparison between in situ observations and ERA5 reanalysis products is given in Figure S1 and Table S1. These model inputs represent averaged values over the selected ERA5 grid cells of about 31 km  $\times$  31 km. ERA5  $T_a$  is on average positively biased during cold and clear-sky conditions, due to a positive bias in snow/ice surface temperature up to 4 °C (Herrmannsdörfer et al., 2023, Fig. S1b). This  $T_a$  positive bias is associated with a moist bias (Fig. S1c). Other reported discrepancies in the ERA5 reanalysis are the underrepresentation of liquid water clouds, with impact on  $Q_s$  and  $Q_t$  (Fig. S1d, e), and a mismatch in the seasonal variation of wind speed (Xi et al., 2024, Fig. S1a). Wagner et al., (2022) estimated total snow accumulation of 98–114 mm in snow water equivalent (SWE) from late October 2019 to early May 2020, which is in line with the ERA5 reanalysis product (115 mm) during the same period (Fig. S1f). Despite these discrepancies with the MOSAiC observations, ERA5 reanalyses were preferred to observations to force the HIGHSTI model because data gaps exist among in-situ measurements, in particular the longwave and shortwave radiative fluxes (Fig. S1d, e).

# 2.2 Snow data




During MOSAiC, comprehensive sea ice and snow observations were carried out (Nicolaus et al., 2022). Snow pit measurements were taken at least weekly but often on several days per week, and occasionally more than once a day. Most measurements were taken within the central observatories in designated clean, undisturbed snow fields. Snow pits were dug at various locations on undeformed first-year ice (107, 142), multi-year ice (20, 76), and places close to open leads (16, 17), ponds (5, 2), and pressure ridges (61, 35). The numbers in parentheses indicate the sample counts for surface and bulk snow, respectively. More than half of the surface and bulk snow samples (51% and 52%) were collected over the first-year ice (FYI). In this study, we only use snow density measurements collected with the 100 cm³ density cutter and the ETH SWE tube. MOSAiC ice floe was snow-free from late June to early July 2020 (Itkin and Liston, 2024), when observations were interrupted due to logistical constraints. The SSL observed over bare ice during the melting season was included among the snow observations.

## 125 **2.2.1** Surface snow density and density profile

The vertical profile of snow density was determined by sampling snow from the vertical wall of snow pits using a 3-cm-high density cutter box, resulting in a vertical resolution of 3 cm for the snow density profiles. Snow density was calculated as the ratio of the weight to the volume of the snow samples. Our analysis mainly focused on the uppermost density measurement in each profile, i.e. the surface snow density ( $\rho_{sfc}$ ). A total of 209 surface snow density samples were collected during the entire MOSAiC, with most samples taken on undeformed sea ice. The uncertainty in these density measurements is about  $\pm$  5-9 % with a tendency to overestimate (underestimate) densities below (above) a threshold between 296 and 350 kg/m<sup>3</sup> (Proksch et al., 2016).

# 2.2.2 Bulk snow density

The bulk snow density ( $\rho_b$ ) can be derived by integrating the measured densities along the vertical profile. However, downscaling high-resolution data to coarser resolution may introduce errors (Fowler et al., 2007). We therefore derived the bulk snow density based on snow depth ( $h_s$ ) and snow water equivalent (SWE) measurements taken once for each snow pit

visit using the ETH-SWE cylinder snow sampler (Macfarlane et al., 2023c). The ETH-SWE tube has a scale on the side to measure snow depth. The bulk snow density  $\rho_b$  is calculated as  $\rho_b = (SWE \times \rho_w)/h_s$ , where  $\rho_w$  is the density of fresh water (1000 kg/m<sup>3</sup>). We analysed 272 bulk snow density samples from snow depth and SWE measurements. The uncertainty associated with density measurement using the SWE tube is approximately  $\pm$  4% (Beaudoin-Galaise and Jutras, 2022). A quality control procedure (Sturm et al., 2002b) was applied to the snow density data. Measurements exceeding 700 kg/m<sup>3</sup> across the entire dataset, as well as those above 500 kg/m<sup>3</sup> recorded from October to May, were excluded.

## 2.3 Snow density parameterisations



In climate models, snow density is either prescribed as a constant or parameterised as bulk values (McCreight and Small, 2014). The parameterisation of bulk snow density is often derived based on a large set of in-situ snow density observations using multiple empirical regression techniques involving a set of proxy variables, such as air temperature, wind speed, snow depth, and age of the snow cover (Pistocchi, 2016; Mizukami and Perica, 2008). In this study, we assess three bulk snow density parameterisations and one prognostic snow density equation (Table 1).

Table 1: Bulk snow density parameterisation schemes.

| Sources                | Snow density scheme                                                                                                                                                                                                                                                                                                                                      |
|------------------------|----------------------------------------------------------------------------------------------------------------------------------------------------------------------------------------------------------------------------------------------------------------------------------------------------------------------------------------------------------|
| Sturm et al., (2010)   | $\rho_b = \rho_0 + (\rho_m - \rho_0) \times [1 - \exp(-k_1 \times h_s - k_2 \times DOY)] $ (E1)                                                                                                                                                                                                                                                          |
| Parameters             | $k_1 = 1.0 \times 10^{-3}$ and $k_2 = 3.8 \times 10^{-3}$ in this scheme are the fitting parameters, $h_s$ (cm) is snow depth, DOY is day-of-year. $\rho_0$ (250 kg/m <sup>3</sup> ) and $\rho_m$ (500 kg/m <sup>3</sup> ) are the fitted baseline and maximum snow densities, respectively, for this study.                                             |
| Bruland et al., (2015) | $\rho_{b} = \rho_{0} + (\rho_{m} - \rho_{0}) \times [1 - \exp(-k_{1} \times h_{s} - k_{2} \times z - k_{3} \times T_{d} - k_{4} \times V^{*})] $ (E2)                                                                                                                                                                                                    |
| Parameters             | $k_1=5.03\times 10^{-3}$ , $k_2=1.8\times 10^{-4}$ , $k_3=4.77\times 10^{-3}$ , $k_4=4.2\times 10^{-4}$ ; $h_s$ (cm) is snow depth, $z$ (m) is the elevation or the height above sea level of the location of the measurement; $T_d$ (°C-days) is accumulated positive degree-day; $V^*$ (m/s-days) is wind days when $T<0$ °C and wind speed $V>2$ m/s. |

| Szeitz and Moore (2023) | $\rho_b = \rho_o + (\rho_m - \rho_0) \times (k_2 \times DOY^2 - \exp(-k_1 \times h_s - k_3 \times T_d - k_4 \times T_{min} - k_5 \times P))$ (E3)                                                                                                                                               |  |  |  |  |  |
|-------------------------|-------------------------------------------------------------------------------------------------------------------------------------------------------------------------------------------------------------------------------------------------------------------------------------------------|--|--|--|--|--|
| Parameters              | $k_1=2.26\times 10^{-1}, k_2=2.29\times 10^{-5}, k_3=1.11\times 10^{-2}, k_4=3.23\times 10^{-4}, k_5=1.96\times 10^{-2}, h_s$ (cm) is snow depth; $T_d$ (°C-days) is accumulated positive degree-day; $T_{min}$ (°C) is the minimum air temperature, $P$ (mm) is the daily total precipitation. |  |  |  |  |  |
| Anderson (1976)         | $\frac{1}{\rho_b} \times \frac{\partial \rho_b}{\partial t} = C_1 \times \exp\left[-d \times \left(T_f - T\right)\right] \times W_S \times \exp\left(-C_2 \times \rho_b\right) $ (E4)                                                                                                           |  |  |  |  |  |
| Parameters              | $T_f$ is the freezing temperature (273.15K); $T$ is the air temperature(K); $W_s$ (m) is the total snow water equivalent; $C_1$ = 5 cm <sup>-1</sup> hr <sup>-1</sup> ; $C_2$ =21 m <sup>3</sup> Mg <sup>-1</sup> and d = 0.08K <sup>-1</sup> .                                                 |  |  |  |  |  |

#### 150 2.4 Snow and ice model




A single-column high-resolution thermodynamic snow and ice model (HIGHTSI) is used to simulate the sensitivity of snow density on the thermal regime and mass balance of snow and ice during the winter-spring period (28 October 2019 - 6 May 2020) when the air temperature was below zero degrees and snow was dry. HIGHTSI computes the energy and mass balances at the snow surface, at the snow/ice interface, within the snow and ice layers, and at the ice bottom (Launiainen and Cheng, 1998). The snow-to-ice transformation is calculated in terms of ice mass balance. The refreezing of slush to snow ice due to ice surface flooding and the refreezing of melted snow to superimposed ice at the snow-ice interface is considered in the model (Cheng et al., 2003, 2008). Internal melting within snow and ice can be modelled (Zhao et al., 2022). HIGHTSI uses timedependent snow density (scheme E4) and snow heat conductivity (Sturm et al., 1997). The thermal properties (density, specific heat, and thermal conductivity) of sea ice are parameterised according to Yen (1981) and Pringle et al., (2007). In this study, we incorporate a time series of MOSAiC surface ( $\rho_{sfc}$ ) and bulk ( $\rho_b$ ) snow density, along with snow density schemes (E1–E4), into the HIGHTSI model to investigate how snow density affects the thermal regime of snow and sea ice (Group 1 of model experiments). The time series of  $\rho_{\rm sfc}$  and  $\rho_{\rm b}$  are presented as 10-day moving averages with temporal standard deviations (c.f. Sect. 3.4). For E1-E3 simulations, the baseline snow density was 250 kg/m<sup>3</sup>. The E4 simulations, which use specified initial snow densities, are labelled as E4(150), E4(250), and E4(320), respectively. For comparison, we performed another group of model experiments (Group 2) using constant snow density values of 180, 200, 220, 250, 270, 300, and 320 kg/m<sup>3</sup>. The initial snow depth and ice thickness were assumed to be 0.1 m and 0.36 m, respectively, representing the mean values for undeformed FYI in the MOSAiC Distributed Network (DN) at the start of the MOSAiC ice camp (Lei et al., 2021). The applied model parameters are summarised in Table S2. Precipitation is one of the most important input variables for snow and ice modelling.

The snow precipitation (expressed in SWE) needs to be converted to snow depth (m) as model input. A density of 340 kg/m<sup>3</sup> (denoted as ρ<sub>s0</sub> in Table S2) was used for the conversion based on mass conservation. For the snow layer on sea ice, snow density is either derived from MOSAiC observations or parameterised.

## 3 Results





#### 3.1 MOSAiC observed snow density statistics

Throughout the MOSAiC expedition, snow density exhibited substantial variability. The values ranged from 82 to 498 kg/m<sup>3</sup> for surface snow and 83 to 690 kg/m<sup>3</sup> for bulk snow. The mean and standard deviation values were 311 kg/m<sup>3</sup>  $\pm$  94 kg/m<sup>3</sup> and 291  $\pm$  106 kg/m<sup>3</sup> for surface and bulk snow, respectively. In the winter-spring period, the surface snow and the bulk snow exhibited the same range of snow density variability (from 82 to 432 kg/m<sup>3</sup> for surface snow and 83 to 433 kg/m<sup>3</sup> for bulk snow), but on average, snow was denser at the surface, with mean  $\pm$  standard deviation values of 283 kg/m<sup>3</sup>  $\pm$  77 kg/m<sup>3</sup> and 253 $\pm$ 59 kg/m<sup>3</sup> for surface and bulk snow, respectively.

Figure 2 presents the frequency distribution of both surface and bulk snow densities. In the winter-spring period, surface snow density peaks at 240 and 310 kg/m³ (Fig. 2a). After snowfall, surface snow undergoes densification due to gravity, wind-induced processes, and temperature-driven metamorphism. The wind exerts a strong effect on the surface snow layer, leading to the formation of wind slab snow (Seligman, 1936) with densities varying from 350 kg/m³ to 500 kg/m³ (Derksen et al., 2014; Domine et al., 2007). For the entire MOSAiC period, the surface snow density distribution (blue line in Fig. 2a) resembles that observed during the winter-spring period. However, the peak is somewhat less pronounced, and the distribution includes a much higher occurrence of large snow densities due to the presence of SSL during the melting period, which had a large weight on the density average (113 surface density samples out of 209 were collected between July and September, when SSL was dominant, Fig. 4e). Most bulk snow density values observed over the winter-spring and the entire MOSAiC periods (Fig. 2b) are around 250 kg/m³ (the mode being 235 kg/m³ and 276 kg/m³ for the winter-spring and entire MOSAiC periods, respectively), in line with the baseline snow density applied in snow density schemes (Table 1). The larger right tail of the bulk snow density distribution compared to the surface snow density distribution is caused by the large summer SSL density values measured close to the ice interface.

Figure 2: Frequency (%) distribution of surface snow density (a) and bulk snow density (b). The blue and red colours represent the distribution for the entire MOSAiC period and the winter-spring period, respectively. N is the sample number. Density bins of 21 kg/m³ and 32 kg/m³ are used in (a) and (b), respectively.

Figure 3: Scatterplot of SWE versus snow depth. The slopes of the linear regression lines correspond to the bulk snow densities for the entire MOSAiC period (blue), the winter-spring period (red), and the MOSAiC legs 4 and 5 (black).

The relationship between SWE (mm) and snow depth (m) is illustrated in Figure 3. The regression slopes represent mean snow densities for each period: 348 kg/m³ (entire MOSAiC campaign), 308 kg/m³ (winter-spring period), and 487 kg/m³ (MOSAiC legs 4 and 5). These values likely reflect the characteristic snow or surface scattering layer densities at a specified period.

# 3.2 Snow density evolution over the annual cycle

The time series of observed surface snow density (ρ<sub>sfc</sub>), bulk snow density (ρ<sub>b</sub>), air temperature, wind speed, snow depth, sampling distribution, and monthly accumulated precipitation are shown in Fig. 4. According to the annual cycle of air temperature, we categorise the periods into four stages. The cold season (defined as Stage I) starts from the beginning of the observations and lasts until 18 February. During this period, the mean air temperature was -25 °C, ranging between -7 °C and -38 °C. The warming period (Stage II) lasts from February 19 to May 10. Within this period, the air temperature increases from -40 °C to -10 °C. The temperature fluctuation between 16 February and 3 March (Fig. 4b) is associated with cyclone passages (Aue et al., 2023). For clarity, we ultimately divided Stages I and II to align with MOSAiC legs 2 and 3. This separation of Stages I and II did not impact our data analyses. The melting period (Stage III) lasts from 13 June until the end of August, with a mean air temperature of 0.1 °C. The re-freezing season (Stage IV) starts from 2 September and lasts until the end of the MOSAiC ice camp. During this period, the mean air temperature was -3.1 °C.

Figure 4: Time series of (a) observed snow density (kg/m³) for surface snow (red) and bulk snow (black); Observed daily average (b) 2-m air temperature (°C) and (c) 10-m wind speed (m/s). The coloured horizontal bars in (a) refer to the MOSAiC legs 1 - 5. The coloured horizontal bars in (b) represent the four stages (dark red: I; green: II; blue: III; and black: IV). (d) Snow depth observations from snow pits. (e) total number of samples per month for bulk snow (black bars) and surface snow (red bars), and (f) observed monthly accumulated precipitation. The values between October and May were measured using vertically pointing 35-GHz Doppler radar (range gate height of 0.23 km), and the rest were observed by a laser disdrometer onboard RV *Polarstern*. (Matrosov et al., 2022).

During Stage I, both surface and bulk snow densities range between 150 and 400 kg/m³, with mean values of 252 kg/m³ for surface snow density and 247 kg/m³ for bulk snow density. In Stage II, the rise in air temperature affects the surface snow density, resulting in a higher surface snow density compared to bulk density. Stage III is characterized by substantial snow and ice melt. During Stage IV, air temperatures dropped below 0°C. This phase is marked by the refreezing of melt ponds and the onset of new snowfall, signalling the beginning of the next winter season. The average wind speed during the study period was 5.9 m/s, with the highest and lowest daily mean values of 13.8 m/s and 1.3 m/s, respectively. The snow depth exhibited significant temporospatial variations (Fig. 4d). The mean snow depth obtained from snow pit observations during the entire MOSAiC period was 0.16 m, with the thickest and thinnest snowpacks measuring 0.49 m and 0.05 m, respectively. This agrees with snow depth observations made with Magnaprobe over several transects in the MOSAiC Central Observatory (Itkin et al., 2024). Snow sampling during the MOSAiC expedition was not evenly distributed in time (as seen by the temporal spread of data points in Fig. 4e, Fig. S2) due to logistical challenges and limited manpower (Itkin et al., 2024), which impacts the snow depth and density distributions.

Figure 5: Simultaneous surface and bulk snow densities extracted from data samples (Fig. 4) during the four stages. The asterisks indicate the mean snow densities (Stage I:  $\rho_{sfc}$  =252 kg/m³,  $\rho_b$  = 247 kg/m³; Stage II:  $\rho_{sfc}$  =319 kg/m³,  $\rho_b$  = 280 kg/m³; Stage III:  $\rho_{sfc}$  =384 kg/m³,  $\rho_b$  = 503 kg/m³, and Stage IV:  $\rho_{sfc}$  =271 kg/m³,  $\rho_b$  = 316 kg/m³). The ellipse shadings represent bivariate Gaussian ellipses in terms of the 95% confidence intervals for each stage. Bar plots on the top and right side depict the absolute frequency (counts) of the observed surface snow density and bulk snow density.

Figure 5 illustrates the relationship between the bulk snow density and surface snow density across four temperature-based stages (I-IV). The bulk and surface snow density samples are collected simultaneously for comparison. The scatter plot displays stage-specific distributions, with ellipses representing correlation patterns. Marginal histograms provide the corresponding frequency distributions. In Stage I and Stage II, a positive relationship exists between bulk and surface snow densities. The ellipse for Stage II is the most flattened, indicating the strongest correlation, whereas Stage I exhibits a weaker correlation with a less flattened ellipse. The slope of the major axis is less than 1, indicating that surface snow density varies more than bulk density. Frequency distributions reveal that bulk snow density is more concentrated, indicating greater uniformity in deeper snow layers. In Stage III, the weakest correlation was observed, with a nearly circular ellipse and a negative slope, indicating

a more independent relationship between the bulk and surface densities. The larger uniformity of the snow samples and smaller spread of the density values in the melting stage, compared to the other stages, probably contributed to this weak relationship, as only melting, heavily metamorphosed snow crystals were observed. In Stage IV, the ellipse is less elongated, and the slope exceeds 1, indicating that bulk snow density varies more than surface snow density. This is coherent with the fact that this stage was characterised by strong spatial variability in snow density, with drifting fresh snow accumulating more in depressions than over dunes. As a result, the frequency distribution shows a broader spread in the bulk than in the surface density.

Figure 6: Normalised mean snow density profiles during each stage. The mean snow depths were 0.15 m, 0.18 m, 0.15 m, and 0.06 in Stage I, II, III, and IV, respectively.

The normalised mean vertical profile of snow density is shown in Fig. 6. Normalisation was necessary because the snow depths at each snow pit differed from one another. During Stage I, the mean snow density gradually increases with depth, rising from 265 kg/m³ at the surface to 310 kg/m³ at the bottom. This is likely because the high-density SSL (snow/ice layer), formed during the melting, remains buried under the new snow at the bottom of the snowpack during the freeze-up. During Stage II, the density distribution reversed compared to Stage I. The snow density gradually decreases with depth, from 310 kg/m³ at the surface layer to 270 kg/m³ at the snow/ice interface. The increase in surface snow density during Stage II compared to Stage I could be attributed to wind compaction (Leeuw et al., 2023), while the decrease in snow density at the snow/ice interface is likely a result of the metamorphism of snow crystals into larger and looser depth hoar (King et al., 2020). At the end of April, inverse temperature gradients occurred in the snowpack (surface temperature became warmer than deeper layer temperatures) associated with warm-air intrusions (Svensson et al., 2023). Density maxima below the surface layer could be a result of fresh snow and an underlying wind-packed layer in some samples.

During the melting in Stage III, two key processes occur that influence the measured density profiles. First, the snowpack becomes saturated with water, resulting in wet, melting snow with a high liquid water content and increased density, especially at the snow-ice interface. Once this wet snow has almost fully melted and surface water is drained away into the ice interior or to neighbouring melt ponds, the second phase begins: the formation of the SSL. The SSL forms through the melting of preferential ice crystal boundaries. This phase also has its highest density near the ice-SSL interface, gradually decreasing toward the surface where melting is more advanced. Hence, while the snow during Stage III is indeed denser than the snow present in earlier stages, it is important to recognise that this density profile develops in two stages, initially from saturated, dense snow, and later from the emerging SSL with its vertical density gradient. During the sea-ice refreezing in Stage IV, the density profile includes the SSL at the bottom and successively accumulated snow layers on the top. Hence, the density profile shows a linear increase with depth, similar to the vertical profile in Stage I. As a result, the observed snow density at each vertical level exhibits distinct spatial variability (Fig. S3).

The mean density profiles shown in Fig. 6 were based on snow samples collected by horizontally pushing the snow cutter in the snow pit wall. The mean bulk density obtained from these profiles (274 kg/m³, 302 kg/m³, 421 kg/m³, and 287 kg/m³ during stage I, II, III, and IV, respectively) should be comparable to the mean bulk density obtained from the SWE tube data (247 kg/m³, 280 kg/m³, 503 kg/m³, and 316 kg/m³ during stage I, II, III, and IV, respectively) presented in Figs. 2-5. For stages I, II and IV, the difference between the mean densities based on the snow cutter and the SWE tube is at the limit of the uncertainty margin. However, during stage III, when liquid water and, later, SSL were present, the difference is more substantial, as the snow cutter data provide a density value that is 20% lower than the data based on the SWE tube. We argue that, in the presence of liquid water at the snow-ice interface, the snow samples collected with the box cutter only reached the top of the water-saturated snow layer, while the SWE tube penetrated through the water-saturated snow layer. Hence, in these conditions, the bulk snow density measured with the box cutter only includes the upper and less dense portion of the snow column sampled with the SWE tube.

## 3.3 Impact of wind on snow density




Wind compacts the snowpack, increasing its density. Strong winds can also cause drifting and blowing snow, leading to significant spatial variability in snow depth, which in turn affects snow density. To quantify wind-induced snow compaction, linear regressions were performed using cumulative wind speed as the predictor. Cumulative wind speed was defined as the sum of daily mean wind speeds (in m/s) over a specified number of consecutive days, analogous to the concept of degree-days used for temperature. The analysis focused on the sensitivity of regression coefficients to the length of the cumulative time window, which ranged from 3 to 30 days.

Although the correlation values are low, all regression coefficients presented in Fig. 7 are statistically significant at the 95% confidence level, with those corresponding to the 3-day accumulation period also significant at the 99% level. The regression

coefficient for surface snow density decreases from 1.5 for a 3-day accumulation window to 0.7 at 20 days, remaining nearly constant thereafter (Fig. 7a). This indicates that the sensitivity of surface snow density to cumulative wind speed is highest over short periods but diminishes as the accumulation period increases. In contrast, the regression coefficients for bulk snow density are consistently lower than those for surface snow density, ranging from 1.0 for a 3-day window to 0.2 for a 30-day window (Fig. 7a, b). This difference is expected, as wind-induced momentum flux most efficiently deforms snow crystals at the snowpack surface.

Figure 7: (a) Linear regression coefficients for the dependence of the surface (red) and bulk (black) snow density on accumulated wind speed, with the accumulation time window ranging from 3 to 30 days. The standard error of each regression coefficient is marked by an error bar. (b). Example for 3 days wind speed accumulation versus surface(red) and bulk(black) snow densities. The solid lines represent the ordinary least squares linear regression fits.

The observed decrease in regression coefficients with longer accumulation periods (Fig. 7a) may be interpreted as follows. After a snowfall event, wind can rapidly deform newly deposited snow through mechanical processes such as particle impact, shear stress, and saltation. The loosely bonded, fragile crystals at the snowpack surface are particularly susceptible to breakage under wind-induced momentum flux, resulting in a rapid increase in snow density (Schmidt, 1980). However, as these weak crystal structures are compacted and broken down, the surface layer becomes denser and more cohesive, reducing its liability to further mechanical deformation. Consequently, the rate of wind-induced densification slows over time, and thermally driven metamorphism begins to play a more dominant role in snowpack evolution (Sturm and Benson, 2004).

## 320 3.4 Evaluation of snow density parameterisations





Figure 8 shows the observed 10-day moving average and parameterised snow densities. In November, the variations in surface and bulk snow densities are opposite. From December to April, both surface and bulk snow densities increased. Overall, the correlation between the surface and bulk densities is 0.73, and the surface snow density has a larger standard deviation than the bulk snow density. In late April to early May, the surface snow density revealed a large drop from 320 kg/m³ to 190 kg/m³ with a standard deviation of 48 kg/m³. This change may be attributed to the abundant precipitation during spring months (Matrosov et al., 2022). E1 scheme, which considers both snow depth and the number of winter days (DOY) as influencing factors, shows a gradual increase in snow density over time. The simulated values agree with the observed densities in the lower to mid-range. However, scheme E1 struggles to represent the response of density to large changes in snow depth. For instance, the effects of heavy snowfall events in November and February (visible in Fig. 4d) are characterised by short-term rapid increases in snow depth, as seen in the observed bulk density changes (Fig. 4a and 8), which are not reflected in the E1 parameterisation. Consequently, E1 produces overly smooth snow density evolution, missing the pronounced effects of snowfall events.

In comparison, scheme E2, which considers the effects of snow depth, positive degree days, and wind speed days, reproduces density changes in response to snow depth changes in November and February. However, E2 consistently overestimates the bulk snow density relative to the observations.

Scheme E3, which depends on snow depth, temperature, accumulated positive degree-days, minimum air temperature, and precipitation, accounts for both mechanical and thermodynamic processes that influence snow structure and density. However, in November, E3 shows an excessively strong increase in snow density compared to the observations, and it simulates a too slow increase of snow density until March. This indicates that, in E3, snow depth plays a crucial role in controlling snow

density early in the season, but other factors, such as the duration of winter, may become more influential later in the season. For E1-E3, initial densities were set below baseline values due to environmental temperature and day of year (DOY) effects.



The snow density scheme E4 is the most complex one out of the four schemes evaluated here, and it is widely used in the sea ice research community (e.g., Essery et al., 2013). The scheme uses a prognostic equation that considers factors such as compaction due to overlying snow layers and metamorphic processes driven by temperature gradients. The initial snow density needs to be specified depending on the application. Here, we select three categories of initial snow density to represent fresh new snow (150 kg/m³), intermediate snow (250 kg/m³), and average snow on Arctic sea ice (320 kg/m³) (King et al., 2020). The E4 simulations highlight the strong impact of initial snow density on snowpack evolution, especially early in the season. The lowest initial snow density results in the lowest predicted snow density throughout the season and provides a lower-bound estimate for snow density. It agrees with the observed lower values of density, especially early in Stage I. The intermediate initial condition gives a moderate estimate of snow density, generally following the observed evolution, but staying below the highest observed density values. Those can only be simulated with the highest initial density values. Although there is an obvious difference between the three different initial snow densities (range of 170 kg/m³), the final difference in density at the end of the winter is relatively small (in the order of 15 to 20 kg/m³).

Figure 8: Temporal evolution of 10-day moving average bulk (black) and surface (orange) measured snow densities, along with their standard deviations (dark and light shaded areas for bulk and surface densities, respectively), as well as the time series of four snow density parameterisation schemes E1-E4 (coloured lines). E4 (magenta lines) was initialized with three different densities (150 kg/m³, 250 kg/m³, and 320 kg/m³). All simulations are for the winter-spring period (from October to May), covering stages I (red line at the bottom) and II (blue line at the bottom).

Table 2: Root-mean-squared error (RMSE), mean absolute error (MAE), and mean bias between observed and parameterised (E1-E4) daily mean bulk snow density. The number in parentheses represents the baseline snow density for E1-E3 and the initial snow density for E4.

|           | Daily mean bulk snow density (kg/m³) |         |         |          |          |          |  |
|-----------|--------------------------------------|---------|---------|----------|----------|----------|--|
|           | E1(250)                              | E2(250) | E3(250) | E4 (150) | E4 (250) | E4 (320) |  |
| RMSE      | 39                                   | 75      | 31      | 33       | 55       | 96       |  |
| MAE       | 29                                   | 72      | 25      | 24       | 51       | 94       |  |
| Mean bias | 22                                   | 72      | 9       | 9        | 51       | 93       |  |

The root-mean-squared error (RMSE), mean absolute error (MAE), and mean bias of the parameterised daily mean snow densities are presented in Table 2. All parameterisations (E1-E4) have a positive mean bias of calculated bulk snow density compared to the observations. E2 and E4(320) exhibit the largest RMSE and MAE compared with the observations. E1 and E3 show lower RMSE and MAE, suggesting improved performance in capturing the observed snow density evolution compared to E2. It is important to note that the comparison with observations is based on the daily mean bulk snow density, which may obscure some of the dynamic aspects of the snow density evolution, such as the spatially heterogeneous snow compaction and redistribution driven by the interaction between wind and surface roughness. Daily averaging may reduce the short-term variability in snow density data and diminish the ability to capture short-term changes and localised events within the snowpack. The results of the E4 scheme highlight the importance of the snowpack conditions at the beginning of the season. The initial snow density significantly influences the evolution of the snowpack. The lowest RMSE and MAE are obtained with an initial snow density of 150 kg/m³. None of the snow density schemes can capture the short-term temporal variability of the observed snow density.

## 375 **3.5 Model experiments**




# 3.5.1 Sensitivity of sea ice to snow density

The modelled mean temperatures of the snow surface ( $T_{sfc}$ ), snow layer ( $T_{snow}$ ), and ice layer ( $T_{ice}$ ), as well as the net increase in ice thickness ( $H_{ice}$ ) during the simulation period (from 28 October 2019 to 6 May 2020), are presented in relation to the mean snow density ( $D_s$ ) used in each modelling experiment (Table 3). The mean  $\rho_{sfc}$  is 24 kg/m³ larger than  $\rho_b$ . The mean standard deviations of  $\rho_{sfc}$  and  $\rho_b$  are 47 kg/m³ and 28 kg/m³, respectively. The mean bulk snow density ( $\rho_b$ ) of the E1-E4

simulations is 294 kg/m<sup>3</sup> with a standard deviation of 41 kg/m<sup>3</sup>. The Ds values calculated applying E1, E3, E4(150) and E4(250) fall within the  $\rho_{\rm sfc}$  observation range [231-325 kg/m<sup>3</sup>], with the E1 yielding the closest match to the observed mean  $\rho_{\rm sfc}$  (279 vs. 278 kg/m<sup>3</sup>). On the other hand, the  $D_s$  calculated applying E1, E3, and E4(150) fall within the  $\rho_b$  observation range [226-282 kg/m<sup>3</sup>], with the E4(150) scheme yielding the closest match to the observed mean  $\rho_b$  (265 vs. 254 kg/m<sup>3</sup>).

Table 3: The mean snow density (Ds) and modelled mean values of surface temperature ( $T_{sfc}$ ), snow temperature ( $T_{snow}$ ), ice temperature ( $T_{ice}$ ), and net increase in ice thickness ( $H_{ice}$ ) using the 10-day moving average of observed surface ( $\rho_{sfc}$ ) and bulk ( $\rho_b$ ) snow density, as well as snow density schemes E1-E4 with different initial snow densities. All values represent averages for the period from October to May. The  $\pm$  values indicate the results incorporating the standard deviations of  $\rho_{sfc}$  and  $\rho_b$ , respectively.

| Mean values during simulation period | $ ho_{ m sfc}$                                      | $ ho_{	extsf{b}}$ | E1    | E2      | E3      | E4 (with different initial snow density kg/m³) |       |       |
|--------------------------------------|-----------------------------------------------------|-------------------|-------|---------|---------|------------------------------------------------|-------|-------|
|                                      | 10-day moving average $\rho_0 = 250 \text{ kg/m}^3$ |                   | 1     | E4(150) | E4(250) | E4(320)                                        |       |       |
| D <sub>s</sub> (kg/m <sup>3</sup> )  | $278 \pm 47$                                        | $254 \pm 28$      | 279   | 329     | 239     | 265                                            | 305   | 347   |
| T <sub>sfc</sub> (°C)                | -24.6±0.03                                          | -24.6 ±0.03       | -24.8 | -24.7   | -24.9   | -24.6                                          | -24.6 | -24.5 |
| Tsnow (°C)                           | -18.0± 0.4                                          | $-17.8 \pm 0.3$   | -16.6 | -17.2   | -16.1   | -17.8                                          | -18.2 | -18.6 |
| Tice (°C)                            | -7.3± 0.4                                           | $-7.0 \pm 0.2$    | -5.9  | -6.5    | -5.5    | -7.1                                           | -7.4  | -7.8  |
| H <sub>ice</sub> (m)                 | $1.08 \pm 0.4$                                      | $1.05 \pm 0.03$   | 0.87  | 0.97    | 0.80    | 1.06                                           | 1.10  | 1.16  |

In response to the Ds range of 226–347 kg/m³, the modelled mean surface temperature fluctuated by 0.4°C, showing a weak dependence on the applied snow density scheme (Fig. 9a). The corresponding  $T_{snow}$  and  $T_{ice}$  exhibited ranges of approximately 2.5°C and 2.3°C, respectively, showing a stronger response to snow density compared to the snow surface temperature (note the different vertical axes in Fig. 9a, b, and c). The modelled net increase in ice thickness ( $H_{ice}$ ) was 1.01 ± 0.12 m, with a variation range of 0.36 m. Different snow density schemes resulted in variations in the modelled net increase in ice thickness, with a standard deviation of 0.12 m, representing approximately 12% of the modelled net ice growth.



Linearity of  $T_{sfc}$ ,  $T_{snow}$ ,  $T_{ice}$ , and  $H_{ice}$  with mean  $D_s$  is observed in two clusters of model experiments and observations (cluster 1 including E1, E2 and E3, and cluster 2 including mean  $\rho_{sfc}$ , mean  $\rho_b$ , and the three E4s). The linearity indicates that the modelled  $H_{ice}$  increases with increasing snow density, while the modelled  $T_{snow}$  and  $T_{ice}$  decrease with increasing snow density. A higher snow density results in higher thermal conductivity and volumetric heat capacity, enhancing the conductive heat flux and allowing more heat to be transferred toward the snow surface. This enhanced heat transfer, associated with higher snow density, also accounts for the thicker ice and slightly warmer surface temperature. It is noteworthy that the differences in the modelled  $T_{sfc}$ ,  $T_{snow}$ ,  $T_{ice}$ , and  $H_{ice}$  may be attributed not merely to the differences in snow density alone, but also to other parameters associated with snow density, such as thermal conductivity and volumetric heat capacity. This is because HIGHTSI applies time-dependent snow thermal conductivity, parameterised as a function of snow density (Sturm et al., 1997). This may explain why the  $D_s$  values of the time series for  $\rho_{sfc}$  and E1 are nearly identical, while the calculated snow and ice variables

differ (Table 3). When a constant snow density is applied in HIGHTSI, the linear relationship between snow density and the thermodynamic state of snow and ice becomes more pronounced and aligns with that of cluster 1 model experiments (E1–E3). 

T<sub>sfc</sub> and H<sub>ice</sub> increase linearly by 0.27°C and 0.23 m, respectively, while T<sub>snow</sub> and T<sub>ice</sub> decrease linearly by 1.48°C and 1.34°C, respectively, in response to an increase in snow density from 180 to 320 kg/m³ ("+" connected lines in Fig. 9).

Figure 9: Modelled (a) mean snow surface temperature ( $T_{sfc}$ ), (b) averaged in-snow ( $T_{snow}$ ) and (c) in-ice ( $T_{ice}$ ) temperatures and (d) mean net increase in ice thickness ( $H_{ice}$ ) as a function of the mean snow density ( $D_s$ ). The symbols presented in the picture represent results of modelling group1 using different snow density schemes (E3:  $\circ$ ,  $\rho_{b}$ :  $\Box$ , E4(150):  $\diamond$ ,  $\rho_{sfc}$ :  $\Delta$ , E1:  $\nabla$ , E4(250):  $\star$ , E2:  $\times$  and E4(320): \*). The "+" connected lines represent values obtained from modelling group 2 experiments applying constant snow density values of 180, 200, 220, 250, 270, 300, and 320 kg/m<sup>3</sup>.

# 415 3.5.2 Sensitivity of sea ice to snow precipitation


To investigate the sensitivity of the modelled sea ice to the uncertainty in the snow precipitation, the cumulative precipitation of each month (taken from Table 1 in Matrosov et al. (2021)) during the winter-spring period was evenly distributed to hourly intervals within the month. This even distribution will retain the observed month-to-month precipitation variability but not the day-to-day variability, which we consider not crucial for our sensitivity study. For each of the six precipitation datasets, a model experiment was conducted, numbered according to the order of the precipitation measurement methods listed in Table 1 of Matrosov et al. (2022). The model experiments were carried out in the same manner as in the case of snow density (ERA5 weather forcing, same assessed model parameters). The results are presented in Fig. 10 and Table 4.

Figure 10: Modelled (a) mean snow surface temperature ( $T_{sfc}$ ), (b) averaged in-snow ( $T_{snow}$ ) and (c) in-ice ( $T_{ice}$ ) temperatures, and (d) net increase in ice thickness ( $H_{ice}$ ) as a function of the measured total accumulated precipitation (expressed in SWE) obtained from the six measurement methods applied in Table 4.

Table 4: Modelled mean snow and ice thermodynamic parameters using snow precipitation (SWE) accumulated during the modelling period. Model runs P1-P6 were carried out using the monthly snow accumulation data listed in Table 1 of Matrosov et al. (2022), which were obtained with different sensors: the vertically pointing 35-GHz Doppler radar (KAZR) with a range gate height of 0.17-km AGL onboard the RV Polarstern (Run P1), the Present Weather Detector (PWD) onboard the RV Polarstern (Run P2), the PWD at the CO ice camp (Run P3), the Pluvio precipitation gauge at the CO ice camp (Run P4), the Particle Size and Velocity (PARSIVEL-2) optical disdrometer onboard the RV Polarstern (Run P5), and the KAZR with a range gate height of 0.23-km AGL onboard the RV Polarstern (Run P6).

|                        | Run P1 | Run P2 | Run P3 | Run P4 | Run P5 | Run P6 |
|------------------------|--------|--------|--------|--------|--------|--------|
| SWE (mm)               | 96     | 81     | 160    | 218    | 53     | 108    |
| T <sub>sfc</sub> (°C)  | -24.9  | -24.9  | -25.1  | -24.9  | -24.7  | -24.9  |
| T <sub>snow</sub> (°C) | -15.8  | -16.1  | -15.2  | -15.6  | -17.1  | -15.7  |
| T <sub>ice</sub> (°C)  | -5.3   | -5.6   | -4.6   | -5.0   | -6.4   | -5.1   |
| $H_{snow}$ (m)         | 0.30   | 0.25   | 0.43   | 0.54   | 0.16   | 0.34   |
| $H_{ice}$ (m)          | 0.76   | 0.81   | 0.64   | 0.71   | 0.96   | 0.73   |


- The modelled surface temperature was not sensitive to the uncertainty in the total snow accumulation, as shown by the small standard deviation (0.13 °C, calculated based on Table 4) between the mean values obtained applying different precipitation data. The standard deviation of the modelled mean in-snow and in-ice temperatures (both 0.6 °C, calculated from Table 4) reveals that the modelled temperature response to uncertainties in snow precipitation is somewhat weaker than the response to the uncertainty in parameterised snow density (std values of 1.0 and 0.9 °C for  $T_{snow}$  and  $T_{ice}$ , respectively, calculated from Table 3). The modelled  $T_{snow}$  and  $T_{ice}$  both vary within a range of 2.0 °C for different precipitation forcing (Figure 10 b, c, Table 4), which is slightly less than the range of  $T_{snow}$  and  $T_{ice}$  variations with respect to different snow density parameterisations (within 2.5 °C). The sensitivity of modelled  $T_{snow}$  and  $T_{ice}$  to the applied precipitation forcing (standard deviation of 0.11 m calculated from Table 4) was close to the model sensitivity to the snow density parameterisation applied (standard deviation 0.14 m calculated from Table 3).
- The modelled snow and ice temperatures, as well as the snow and ice thicknesses, show a nonlinear relationship to SWE during the modelling period. More snow accumulation leads to a thicker snowpack and a stronger insulation effect. As a result, the in-snow and in-ice temperatures increase, resulting in thinner ice. The response of the modelled parameters (Fig. 10) to increased SWE is opposite to their response to increased snow density. However, a further increase in snow accumulation would reverse the above processes due to snow-to-ice transformation. For example, runs P3 and P4 yielded a net of 0.08 m and 0.15 m snow-ice, respectively, because the initial ice thickness was relatively thin (see Table S2) and the accumulated precipitation was large (see Table 4), the net increase in ice thickness (run P4) began to increase (Fig. 10d).

## 4 Discussion




The mean surface/bulk snow densities are comparable during stages I (252/247 kg/m³), II (319/280 kg/m³) and IV (271/316 kg/m³), as the bulk snow density primarily reflects the cumulative effect of overlapping surface snow layers. These values are also consistent with Northern Canadian Arctic snow densities, which range from 250 to 350 kg/m³ (Bilello, 1967; Zhao, et al., 2023), indicating a long-term consistency in snow properties across the Arctic over decades. The SWE-snow depth regression using all data samples yielded an overall mean snow density of 348 ± 9.6 kg/m³. This value agrees well with the mean bulk snow density (340 kg/m³) observed during the SHEBA expedition (Sturm et al., 2002b) and is comparable to the mean snow density of 339 kg/m³ observed over the Antarctic sea-ice (Massom et al., 2001), and specifically to the mean densities of 282 and 356 kg/m³ observed in the Weddell Sea over first-year and second-year ice, respectively (Nicolaus et al., 2009). During the melting season (III), the bulk snow (SSL) density exceeds the surface snow (SSL) density by 158±150 kg/m³. Before the meltwater drainage, the bottom layer of the snowpack densifies by becoming soaked with meltwater. After the drainage, the intense surface melting and sublimation reduce near-surface SSL density, while at deeper layers, the ice disintegration into SSL is less advanced and the density of the SSL is closer to the density of the ice from which it originates (Macarlane et al., 2023c). As a result, SSL density increases almost monotonically from the surface to the bottom.

During winter, wind enhances spatial heterogeneity in snow depth and density through interactions with surface topography. At the MOSAiC Central Observatory, Itkin et al., (2023) demonstrated that the interaction between drifting snow and sea ice roughness explains up to 85% of the observed snow depth variability over both level and deformed ice. In the Canadian Arctic, Iacozza and Barber (1999) observed that the wind direction during depositional storm events impacts the distribution of snow dunes. During the MOSAiC winter-spring period, the signature of the wind-snow interaction is seen in the peak of the surface and bulk snow density distributions around 250 kg/m³ (red dotted line in Figure 2a and 2b). The distribution of surface snow density during the entire MOSAiC period shows a larger fraction of dense snow, likely due to the contribution of wet snow and the summer SSL.



The mean vertical profile of snow density exhibited distinct patterns at each stage, with temporal variations in snow density being more pronounced than vertical variations (Fig. 6). Both surface and bulk snow densities exhibited a non-monotonic increase during the winter and a decrease by the end of spring (Fig. 8). The increase in snow density during winter aligns with Arctic snow climatology (Warren et al., 1999). However, the decrease in snow density during the melting in May is consistent with what has been observed also over Antarctic sea-ice (Nicolaus et al., 2009). Warren et al., (1999) concluded that the average Arctic snow density could reach 320 kg/m³ in May after autumn and winter settling and wind packing.

Finding a robust relationship between bulk snow density samples and air temperature using MOSAiC data remains challenging because snow density is affected by wind-driven compaction and wind interaction with surface topography. Snow melting induces simultaneous density increases and decreases at different depths in the snpwpack. Similar difficulties in observing a relationship between air temperature and snow density were reported in previous studies conducted in mountainous regions, in the terrestrial Arctic, and over the Greenland ice sheet (Zhao et al., 2023; Howat, 2022). On the other hand, air temperature demonstrates a positive correlation with fresh snow density (Sturm and Holmgren, 1998; Judson and Doesken, 2000). In case of dry snow, air temperature influences the evolution of snow density by the densification process, as it governs the vertical temperature gradients within the snowpack (Zhao et al., 2023). However, in the different stages outlined in this study, the impact of air temperature on snow density becomes evident. In Stage III, when temperatures approached 0°C, the mean SSL bulk density reached 503 kg/m³, showing an increasing density from the surface to the bottom of the SSL layer. This contrasts with the bulk snow density of 316 kg/m³ observed during other stages, when air temperatures were significantly lower.

Snow density increases with increasing accumulated wind speed. Qualitatively similar relationships were detected in previous campaigns over the Arctic Ocean, Greenland, and the Tibetan Plateau (Zhao et al, 2022; 2023; Howat, 2022). Furthermore, the differential sensitivity of surface and bulk snow densities to accumulated wind speed underscores the importance of temporal scales when considering wind-snow interactions. The effect of wind on surface snow density is more direct than the effect of wind to bulk snow density, which agrees with previous work (Meister, 1989; Sokratov and Sato, 2001; Walter et al., 2024).

Snow density schemes simulated the gradual increase in snow density during the winter-spring period (Fig. 8). However, none of the snow density parameterisations adequately captured the observed temporal variability in April - May. The temporal evolution of snow density influences the thermal conductivity and volumetric heat capacity of snow, which in turn affects the thermal inertia and mass balance of sea ice. Some results from the parameterisation schemes fell within the range of the observed values and aligned with the snow density climatology of Warren et al. (1999), suggesting their general applicability for sea ice and climate models.




During the winter-spring period, the mean spatial standard deviation of the surface snow density (77 kg/m³) was larger than that of the bulk snow density (59 kg/m³). This illustrates the strong spatial variability of the surface snow properties (Fig. 8). King et al., (2020) observed that snow density is highest in thin snow layers over undeformed ice and lowest in thicker snow layers over older and deformed ice. Higher densities over thin snow layers are due to the stronger wind compaction over smooth ice, while thicker snow is less dense because of the loose depth hoar bottom (Wagner et al., 2022). This depth hoar is caused by metamorphism occurring when the snowpack is exposed to temperature gradients (King et al., 2020).

The analysis of the sensitivity of sea-ice to snow density showed that the modelled mean surface temperature ( $T_{sfc}$ ), snow temperature ( $T_{snow}$ ), ice temperature ( $T_{lce}$ ), and ice thickness ( $H_{lce}$ ) exhibited linearity with mean snow density when applying snow density schemes E1, E2, and E3 with the same initial snow density (Fig. 9). Scheme E4, with different initial snow densities and time series of surface and bulk snow densities, produced a distinct linear trend in these modelled parameters, with a smaller coefficient of the regression slope. The temperature and ice thickness differences originating from differences in modelled parameters reached 0.2 °C for  $T_{sfc}$ , 1.4 °C for  $T_{snow}$  and  $T_{lce}$ , and 0.21 m for Hice when applying snow density scheme E1 and  $\rho_{sfc}$ , despite having the same average snow density (symbols  $\Delta$  and  $\nabla$  in Fig. 9). The analysis of the sensitivity of sea-ice to snow precipitation, on the other hand, revealed nonlinearity in the modelled snow and ice parameters in response to different monthly cumulative snow precipitation forcings.

During autumn, winter, and spring, rain-on-snow (ROS) events can form hard ice crusts on the snow surface (Rennert et al., 2009). During the MOSAiC campaign, a few melting events associated to warm-air intrusions and ROS occurred in September (Leg 5), increasing surface snow density from 150 kg/m³ to 350 kg/m³ within two days (Stroeve et al., 2022). Between 12 – 22 April, two warm-air intrusion episodes raised near-surface temperatures to near-melting conditions (Svensson et al., 2023), causing an inversion in the snowpack temperature gradients, although no snowpack liquid water was documented during this period.

MOSAiC snow density observations are constrained to a single drift trajectory and the specific environmental conditions of the studied ice floe. However, snow density along the MOSAiC drift trajectory captured both temporal (October-August) and spatial (distributed sampling sites) variability throughout the campaign. The Lagrangian approach—tracking the same ice floe over a period—allowed us to isolate the temporal evolution of snow density under observed meteorological forcing. However,

we should not overemphasise the representativeness of the MOSAiC drift trajectory. To understand the overall snow density evolution along the Transpolar Drift Stream, more in-situ observations are needed.

## 530 5 Conclusion and outlook






Snow density measurements obtained using both the density cutter and ETH SWE tube during the MOSAiC expedition were investigated. The sample mean ( $\pm$  std) surface and bulk snow densities over the entire MOSAiC period were  $311 \pm 94$  kg/m³ and  $291 \pm 106$  kg/m³, respectively. During the winter-spring period, the corresponding values were  $282 \pm 77$  kg/m³ and  $253 \pm 59$  kg/m³, respectively. The total sample averages for surface and bulk snow density were  $311 \pm 94$  kg/m³ and  $291 \pm 106$  kg/m³, respectively. SWE-depth regression yielded snow densities of 348 kg/m³ (entire MOSAiC), 308 kg/m³ (winter-spring), and 487 kg/m³ (melting season; primarily SSL density). Unlike density sample statistics, these values reflect the integrated physical relationship between SWE and snow depth. Cumulative wind exposure increased both surface and bulk snow densities. This wind compaction effect was most pronounced during the initial 3-5 days following snowfall events. The modelled mean snow density and temperature were inversely related, i.e. the higher the mean density was, the lower the mean snow and ice temperatures were. Higher snow density leads to a larger snow thermal conductivity, which enhances heat transfer between the atmosphere and underlying sea ice. The model runs using a constant snow density showed strong linearity in the modelled snow and ice parameters as snow density increased. This linearity was consistent with that observed using schemes E1–E3. The response of the modelled snow and ice parameters to the increase in SWE was opposite to their response to the increase in snow density. Based on error statistics, snow density parameterisations E3 and E4 performed better than the others and are recommended for modelling applications.

Retaining a realistic representation of the small-scale spatial variability of snow properties is, however, important for the simulation of the surface energy budget and the ice thickness: our sensitivity tests demonstrated that the modelled ice thickness varied up to 0.14 m and 0.11 m in response to changes in snow density parameterisation and precipitation input, respectively. Therefore, to improve the HIGHTSI representation of snow spatial variability, the snow surface roughness and the effect of drifting snow on snow depth distribution need to be accounted for. This will enable the model to simulate snow depth and density through probability density functions, which represent their spatial variability, rather than through single values for each time step.

In this study, modelling was conducted exclusively for the winter-spring period because HIGHTSI does not simulate the formation of the surface scattering layer (SSL), which was observed during the MOSAiC melting season. The SSL is a common and widespread feature of Arctic sea ice during the melting season (Smith et al., 2022), but it is currently not represented in any existing sea ice models or is simulated merely as a persistent snow layer at the top of the ice surface. This omission or misrepresentation can potentially lead to significant errors in the ice surface energy and mass budgets. To fully understand the

formation and melting of the SSL, as well as the associated erosion of the ice surface, a dedicated SSL modelling component should be developed.

Data availability: The ERA5 reanalysis products are available through the Copernicus Climate Data Store portal (https://doi.org/10.24381/cds.adbb2d47, 2020). Hersbach et al., Snow observations available through https://doi.pangaea.de/10.1594/PANGAEA.935934; The precipitation data is publicly available online from the ARM: https://adc.arm.gov/discovery/#/results; The and wind temperature data are available from: https://doi.org/10.18739/A2PV6B83F.

Author contributions: This study was devised by Y.C. and A.R. Initial data retrieval and analyses, as well as draft preparation, were performed by Y.C. B.C. conducted modelling, data analyses, and contributed to draft manuscript preparation and editing. R.P., T.V., W.D., and A.R.M. contributed to draft preparation, results analysis, and editing. R.D., A.R.M., S.A., and M.S. contributed to manuscript editing. All authors participated in the interpretation of the results and manuscript revisions.

Competing interests: At least one of the (co-)authors is a member of the editorial board of The Cryosphere. The authors have no other competing interests to declare.

Acknowledgements. The authors thank the two anonymous reviewers and Dr Masashi Niwano, the handling editor, for their valuable comments, which greatly improved the manuscript.

Financial support: Y.C. was funded by the Office of China Postdoctoral Council. Y.C., B.C., R.P., W.D., T.V., and A.R. were funded by the European Union's Horizon 2020 Research and Innovation Programme through the Polar Regions in the Earth 575 System project (PolarRES) under Grant 101003590; S.A. was supported by the Alfred-Wegener-Institut, Helmholtz-Zentrum für Polar- und Meeresforschung, the University of Hamburg, the German Research Foundation's (DFG) projects fAntasie (AR1236/3-1) and SnowCast (AR1236/1-1) within its priority program "Antarctic Research with comparative investigations in the Arctic ice areas" (SPP1158), and the DFG Emmy Noether Programme project SNOWflAke (project number 493362232). A.R.M., R.D. and M.S. were supported by the Swiss Polar Institute (SPI reference DIRCR-2018-003) Funder ID: 580 http://dx.doi.org/10.13039/501100015594. European Union's Horizon 2020 research and innovation program projects ARICE (grant 730965) for berth fees associated with the participation of the DEARice project. WSL Institute for Snow and Avalanche Research SLF. WSL 201812N1678. Funder ID: http://dx.doi.org/10.13039/501100015742 and ARM was funded by the Swiss National Science Foundation (SNSF) project number P500PN 217845. A.R. and W.D. acknowledge funding by the Deutsche Forschungsgemeinschaft (DFG, German Research Foundation)— project 268020496 TRR 172, within the Transregional Collaborative Research Center "ArctiC Amplification: Climate Relevant Atmospheric and SurfaCe Processes, and Feedback Mechanisms (AC)3. R.P. and B.C. were also funded through the Research Council of Finland project IceScales (grant n. 364939).

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
