# Peer review of "Seasonal evolution of snow density and its impact on thermal regime of sea ice during the MOSAiC expedition"

_EGUsphere, 2025_

## Author Response (AR1)

Thank you for handling our manuscript (EGUSPHERE-2025-1164) "Seasonal evolution of snow density and its impact on thermal regime of sea ice during the MOSAiC expedition" by Cheng et al.

Based on comments from you and the other two anonymous reviewers, we have conducted a major revision of the manuscript. Our major revision process can be divided into two stages.

Stage 1: Complete the response letter and carry out the corresponding major revision of the manuscript by the end of June.

Stage 2: Further revision of the manuscript based on additional comments from co-authors on the revised manuscript made during stage 1 by 20 August.

As the manuscript has 10 co-authors and there was a summer break between stages 1 and 2, we spent more time finalising it than originally planned.

Here is a summary of the major revisions we have made:

**1) The manuscript has been restructured to improve clarity and the logical flow of the text:**

- I. Figure 2 in the original submitted manuscript (Chapter 2) was moved to the Results (Chapter 3) in the revised manuscript.
- II. The Chapter 4 "Discussion and Conclusion" in the original submitted manuscript has been separated into Discussion (Chapter 4) and Conclusion and Outlook (Chapter 5) in the revised manuscript.
- III. The surface scatter layer (SSL) is described in the Introduction (Chapter 1)

**2) We have modified the text throughout the entire manuscript: a) to improve the clarity; b) to add the missing information, and c) to correct the errors:**

- a. The importance of the surface snow layer is added in the introduction chapter.
- b. The objectives of this study and implementation strategy have been modified and added in the introduction chapter
- c. The weather data section 2.1 was largely reformulated with new information on in situ weather data and ERA5 reanalysis products.
- d. The snow data section 2.2 was reformulated with more information on the snow sampling procedure and observation uncertainties.
- e. The snow density parameterisations section 2.3 has been revised slightly to improve clarity.
- f. Snow and ice model section 2.4 has new and corrected information on model configurations, such as initial conditions and model parameters (Figure S1, Tables S1, S2).
- g. MOSAiC observed snow density statistics section 3.1 has been largely reformulated. The discussions of SSL have been improved. The discussion on the snow density distribution has been revised.
- h. The text in section 3.2 has been revised with more concrete analyses and discussions, particularly for SSL. We added analyses of the mean snow density differences observed between the snow cutter and the SWE tube.
- i. Text in section 3.3 has been reformulated to align with the new Figure 7.
- j. Section 3.4 has been revised substantially to improve the clarity of the discussion and quantify the results.
- k. The subtitles of 3.5.1 and 3.5.2 have been renamed to "Sensitivity of sea ice to snow density" and "Sensitivity of sea ice to snow precipitation" to better align with the research content. The text in both subsections has been revised to enhance the clarity of the modelling investigation and discussions.

- 1. Discussion chapter 4 is reformulated and substantially revised.
- m. The Conclusion and Outlook chapter 5 is reformulated and substantially revised.

**3) Figures and Tables**

The colour of all figures has been compiled with the TC requirement:

- Figure 1 has been redrawn for better clarity. The figure caption has been revised accordingly.
- Figures 2 and 3 remain unchanged, but their captions and interpretations have been revised
- Figure 4 has been redrawn, adding monthly accumulated precipitation in f). After a close discussion with co-authors, we decided to remove the individual snow depth line from Figure 4d because it is a spot measurement and may not represent the overall snow condition of the Central Observatory (CO) and Distribution Network (DN).
- Figures 5 and 6 have been redrawn to comply with the TC figure colour requirement.
- Figure 7 has been redrawn. During the second revision stage, we identified technical errors in the data sets. However, the revised Figure 7 does not affect the conclusions. The standard error of each regression coefficient is added.
- Figure 8 remains the same.
- Figures 9 and 10 have been redrawn to comply with the TC figure colour requirement.
- Figure S1 was redrawn. The in situ meteorological parameters were added to the figure.
- Figure S3 was redrawn to comply with the TC figure colour requirement
- Table 1: More information is added.
- Technical corrections have been carried out for Tables 2 and 3.
- The caption of Table 4 has been reformulated for better clarity.
- Table S1 has been revised to correct errors and add more information
- Table S2 is created to show the statistics comparison between observed and ERA5 reanalysis modelled weather parameters.

**4) The language corrections:**

The language of the entire manuscript has been reviewed and corrected for typos and grammatical and technical errors. Some sentences have been revised from the original text. Due to the numerous language corrections and updates, all changes are highlighted at the paragraph level to ensure readability of the revised manuscript.

Please find the revised manuscript and updated response letter below. We hope the manuscript meets the scientific requirements of The Cryosphere journal and is suitable for publication.

Thank you for your time and help

Best regards,

Bin Cheng and co-authors

Below is our response to the comments by two reviewers. Please note that the text in blue was the author's response made during the first stage of the revision. The text in red is the update of the response made during the second stage of the revision.

**Comments from Referee 1**

**Review Summary:**

The paper discusses the temporal evolution of snow density on Arctic sea ice during the MOSAiC campaign. The observational data are compared with several snow density parameterizations of varying complexity to assess how well they represent changes in snow density. Modeling results are then used in a sea ice model to calculate snow and ice temperatures as well as sea ice thickness.

The results of the mean snow density calculations show relatively good agreement with observations. However, the temporal variations in snow density are not well captured, which affects the accuracy of the resulting sea ice thickness estimates.

The core idea of the paper is clear, and the findings offer a valuable contribution to the understanding of snow compaction processes on Arctic sea ice. The study is particularly relevant for improving snow representation in climate and remote sensing models.

However, the narrative is at times disorganized, and several sections—particularly the introduction and methods—lack structure, which can confuse the reader. The volume of the manuscript is also quite large. For better readability, the **conclusion** section should be separated from the **discussion** and significantly shortened.

There are also methodological and conceptual issues that require clarification:

• Line 13: It is not totally clear why the focus is on the top 3 cm of the snowpack. It would be helpful to explain this in the introduction, potentially in relation to remote sensing relevance or surface energy balance.

There are several reasons for us to define the top 3 cm of snow as the "surface snow layer" and focus on its changes. 1) The "surface snow layer" typically refers to the uppermost layer of snow that is directly in contact with the atmosphere. This layer is characterised by new snow deposition, the surface radiative and turbulent energy fluxes balance, which results in the variation of surface skin temperature, and high sensitivity to weather conditions (e.g., changes in air temperature and wind). 2) For snow and sea ice process modelling, a surface layer between 1 – 5 cm is usually defined depending on the model's vertical resolution. The HIGHTSI model typically has 10 layers in snow and 20 layers in ice. For a snow depth of 20 cm, the surface layer is accordingly 2 cm. 3) Finally, the MOSAiC in situ snow observations were carried out with a vertical layer increment of 3 cm, as this was the size of the density cutter. Accordingly, the vertical resolution of the dataset is 3 cm.

We have added the following text to explain the surface snow layer more clearly in the introduction:

"Surface snow layer refers to the uppermost layer of snow that is directly in contact with the atmosphere. This layer is characterised by new snow deposition and surface skin temperature varying according to the surface energy balance controlled by the radiative and turbulent fluxes. Hence, the layer is sensitive to variations in the air temperature, wind speed, and surface albedo. The bulk snowpack density, on the other hand, is critical for the sea ice mass balance. Accordingly, we address it separately from the snow surface density to highlight the different roles of the surface layer and the bulk snowpack in the sea-ice simulation."

The following text was given in the revised manuscript (L66-71):

"In this study, we investigate the seasonal evolution of snow density during the Multidisciplinary Drifting Observatory for the Study of Arctic Climate (MOSAiC) expedition. We address snow density for the surface snow layer (defined by the sampling resolution of a density cutter, here the top 3 cm) and the bulk snowpack (entire snow layer). The surface snow layer is characterised by new snow deposition. It is highly sensitive to atmospheric influences, responds more rapidly to variations in the surface energy balance than deeper layers, and influences the surface energy balance through its radiative properties and surface roughness. The bulk snowpack density is critical for sea ice mass balance."

• Lines 37–38: More details on the origin and significance of the snow surface layer (SSL) would strengthen the introduction. Some content from section 3.2 (e.g., lines 222–225) could be relocated here.

Agreed, we moved the surface scattering layer (SSL) description with some additions to the Introduction.

SSL develops when the surface of sea ice melts and the brine channels preferentially melt, revealing a porous structure that resembles snow. It originates from the underlying ice and has no snow included in it. Its isotopic signature is purely from the underlying ice. The preferential melting of ice crystal boundaries produces the SSL (Smith et al., 2022), a porous, snow-like layer where the density increases with depth (Macfarlane et al., 2023a). The SSL is less dense than the sea ice from which it is generated. The surface meltwater originating from the melting SSL drains vertically through the ice column, when the ice porosity allows, and laterally into melt ponds and leads, leaving the SSL relatively dry. As the top SSL melts, there is a simultaneous transformation of the underlying bare ice to the SSL. Thus, the thickness of the SSL remains almost stable throughout the summer.

The following text was given in the revised manuscript (L35-41):

"A surface scattering layer (SSL) forms in summer and persists beyond it, buried below new accumulated snow. SSL visually looks like a snow layer. The isotopic signature of SSL is purely from sea ice. The meltwater drains to melt ponds and leads, leaving the SSL relatively dry. As the top SSL melts, there is a simultaneous transformation of the underlying bare ice to the SSL. Thus, the thickness of the SSL remains almost stable throughout the summer, and its density increases with depth (Macfarlane et al., 2023b). SSL reveals a porous structure that resembles snow (Smith et al., 2022). In winter, the densities of snow and the SSL affect the thermal properties of the snowpack (Macfarlane et al., 2023b). "

• Lines 41–42: The statement should be corrected: snow density and depth together determine snow mass. A better phrasing could be: "Snow density and layer thickness determine the weight of the snowpack on sea ice. When the snow load is sufficient to submerge the ice surface, flooding may occur, leading to slush and snow-ice formation."

We partly replaced the original text with the text suggested by the reviewer: "Snow density and layer thickness determine the weight of the snowpack on sea ice". The rest of the text has been removed because the snow-ice and superimposed ice descriptions have been presented in the previous paragraph.

This statement was trivial and therefore removed in the revised manuscript.

• Line 42–43: It's snow microstructure—not just density—that governs percolation and wicking processes.

We have described the snow-ice and superimposed ice formation elsewhere, so this sentence was removed to avoid repetition.

• Lines 50–51: Consider revising to: "This results in changes to the stratigraphy and usually leads to compaction and densification of snow."

Done.

After snowfall, snow density begins to evolve due to snow metamorphism. This results in changes to the stratigraphy and usually leads to compaction and densification of snow (Bormann et al., 2013; Helfricht et al., 2018; Judson and Doesken, 2000).

The following text was given in the revised manuscript (L47-48):

After snowfall, snow density begins to evolve due to mechanical compaction and snow metamorphism, generally leading to densification of snow (Bormann et al., 2013; Helfricht et al., 2018; Judson and Doesken, 2000).

• Line 59: Thermodynamic metamorphism does not result from densification; rather, densification is one outcome of metamorphism. Additionally, kinetic growth during temperature-gradient metamorphism can inhibit compaction.

We improved the description accordingly

The rearrangement of snow microstructure during snow metamorphism is a significant driver of snowpack density. Sublimation and deposition of water vapor under a temperature gradient can drive kinetic growth of faceted crystals, which may inhibit compaction, whereas other metamorphic processes can enhance bonding and increase snow density.

The following text was given in the revised manuscript (L55-58):

The rearrangement of snow microstructure during snow metamorphism affects snowpack density. Sublimation and deposition of water vapour under a temperature gradient can drive kinetic growth of faceted crystals, which may inhibit compaction, whereas other metamorphic processes can enhance bonding and increase snow density (Jafari et al., 2020; Nicolaus et al., 2022).

• Line 71–72: Did you consider the effect of rain-on-snow (ROS) events on snow density?

The ROS was observed during MOSAiC (Stroeve et al., 2022). The events occurred during MOSAiC leg 5 beyond our modelling period. Therefore, we have not done a concrete investigation of ROS effect on snow density. Nevertheless, we present a qualitative discussion on ROS in the revised manuscript (discussion section):

During autumn, winter, and spring, rain-on-snow (ROS) events can form hard ice crusts on the snow surface (Rennert et al., 2009). During the MOSAiC campaign, a few ROS events occurred in September (Leg 5), increasing surface snow density from 150 kg/m³ to 350 kg/m³ within two days (Stroeve et al., 2022). Between 12 – 22 April, two warm-air intrusion episodes raised near-surface temperatures to near-melting conditions (Svensson et al., 2023), although no ROS events were documented during this period.

The following text was given in the revised manuscript (L519-524):

During autumn, winter, and spring, rain-on-snow (ROS) events can form hard ice crusts on the snow surface (Rennert et al., 2009). During the MOSAiC campaign, a few melting events associated to warm-air intrusions and ROS occurred in September (Leg 5), increasing surface snow density from 150 kg/m³ to 350 kg/m³ within two days (Stroeve et al., 2022). Between 12 – 22 April, two warm-air intrusion episodes raised near-surface temperatures to near-melting conditions (Svensson et al., 2023), causing an inversion in the snowpack temperature gradients, although no snowpack liquid water was documented during this period.

• Line 84: Please remove the thin black lines around the plots in Figure 1 and align them. Also, CO2 and CO3 are difficult to distinguish on the left map—consider improving their labels.

Figure 1 has been redrawn accordingly.

Line 88: What was the temporal resolution of the snow measurements? Snow measurements were recorded at a daily temporal resolution. To ensure a spatial measurement distribution, multiple locations for snowpit time series were set up within and around the central observatory (CO). Snow measurements were conducted daily, rotating through these different locations. Due to the number of different locations, individual snowpit sites were revisited on a weekly or bi-weekly basis.

• Lines 93–94: This sentence can likely be removed to save space.

Agree, and the text has been removed. T

The text in Section 2.1 was substantially reformulated.

• Line 96: Please clearly state that ERA5 reanalysis data were used in the calculations.

We have written in the later part of the original manuscript (L158-L161) on this matter: The meteorological parameters, including wind speed (V), air temperature (Ta), relative humidity (Rh), precipitation (P), as well as shortwave (Qs) and longwave (Ql) radiative fluxes from the ERA5 reanalysis along drift trajectory of MOSAiC CO (Leg 1-3) were used as forcing data for the HIGHTSI model (Fig. S1).

For better clarity, we improved the text here as well.

The entire section 2.1 has been reformulated substantially with more descriptions of the in situ observations and ERA5 reanalysis products. The corresponding parts (Figure S1, Table S2) in the supporting material file have been revised and added accordingly.

• Line 99: Include volume and spatial resolution for the box cutter measurements. How many snow pits were excavated per measurement period (weekly?).

The box cutter has a volume of 100 cm3. Each snow sample has a depth of 3 cm. Line 101: "Snow pits were conducted weekly at various locations on undeformed first-year ice, second-year ice, and places close to open leads and pressure ridges. The majority of measurements were taken within the central observatories in designated clean, undisturbed snow areas. Due to limited manpower, individual snowpit sites were often revisited on a weekly or bi-weekly basis. The snow pits are composed of multiple instruments all measured within a 1m x 1m area. Transects of "quick" snow measurements, including a snow micro penetrometer, allowed for spatial quantification of the snow properties. However, in this study, we only use data from the traditional snowpit point measurements for density (100 cm3 density cutter and the ETH SWE tube)

We have made substantial revisions to Section 2.2, 'Snow Data,' in the revised manuscript (L115–139)

• Line 110: Please use the International Classification for Seasonal Snow on the Ground (Fierz et al., 2009) to describe snow types for consistency and clarity.

These snow types are presented only for descriptive purposes. We have therefore removed them from the text.

- Line 111: Clarify what is meant by terms like "drifted snow," "frozen snow," and "jewel snow." For example, is "drifted snow" a wind slab? "Visible dust" is not a snow type.
- These snow types are presented only for descriptive purposes. We have therefore removed them from the text.
- Lines 124–125: Please provide more detail on snow depth measurements.

We derived the bulk snow density based on snow depth and SWE measurements, measured once for each snowpit visit. The ETH-SWE tube has a scale on the side to identify the snow height that is being collected and measured. This is confirmed with a corresponding snow height measurements using a ruler placed vertically against the snowpit wall once it had been excavated.

We have made substantial revisions to Section 2.2, 'Snow Data,' in the revised manuscript (L115–139)

• Lines 127–129: Consider moving this content to the Results section, as it presents findings rather than methods.

Done

• Line 152: Clarify whether this variable was included in the model.

Yes, this process has been taken into account in the HIGHTSI model. We have improved the text accordingly.

• Line 163: Snow density was measured weekly, so how was the 10-day moving average calculated from these discrete in situ observations?

Snow density was measured daily, not weekly. We have corrected this description. We calculated 10-day moving averages applying the actual daily snow density measurements. For example, the first 10-day time window is day 1 - day 10, the next 10-day time window is day 2 - day 11, and so on.

- Lines 166–168: This content might be more appropriate in section 2.2 on snow density. Done, we moved text to section 2.2 accordingly.
- Line 183–184: Gravity?

Added

After snowfall, surface snow undergoes densification due to gravity, wind-induced compaction, and temperature-driven metamorphism.

• Line 214: Figure 4 needs revision. For example, panel (c) lacks a color bar, which is mentioned in the caption. Including precipitation data would also enhance the figure.

Sorry, there was a spelling mistake in the legend of Fig.4. Panel (c) contains just a black line and doesn't need a color bar. But, Panel (a) has the bottom-color bars for legs, and Panel (b) has the bottom-color for the stages. We corrected the legend "The bottom-colored bars in (a) refer to the MOSAiC legs 1-5". We have also modified this figure, and the precipitation subplot has been added.

Figure 4 was redrawn. The figure caption has been revised.

• Lines 222–225: This paragraph would fit better in the Introduction when the SSL is first mentioned.

Done, we shortened and moved the text into the introduction chapter:

SSL develops when the surface of sea ice melts and the brine channels preferentially melt, revealing a porous structure that resembles snow (Smith et al., 2022). Its isotopic signature is purely from the underlying ice. The density of this porous, snow-like layer increases with depth (Macfarlane et al., 2023). The SSL is less dense than the sea ice from which it is generated, but it is much denser than snow. The meltwater drains to melt ponds and leads, leaving the SSL relatively dry. As the top SSL melts, there is a simultaneous transformation of the underlying bare ice to the SSL. Thus, the thickness of the SSL remains almost stable throughout the summer (Macfarlane et al., 2023). SSL visually looks like a snow layer. It has a surface albedo greater than that of bare ice but less than that of snow.

The description of SSL was placed in the revised manuscript (L35 - 41).

• Line 240–241: Improve the readability of the bar plots by standardizing bar sizes and labels. Done. The figure has been revised.

The figure has been revised with a user-friendly colour.

• Lines 251–253: The contribution of the top 3 cm increases as snow depth decreases. However, no correlation between surface snow and bulk density is observed, which is intriguing. This point could be emphasized more clearly.

The lack of correlation between surface density and bulk density during the melting period (Stage III) suggests that SSL evolution is governed by different processes than those driving snow evolution during the dry period. We have addressed this issue in the revised manuscript.

The following text was given in the revised manuscript (L244-247):

In Stage III, the weakest correlation was observed, with a nearly circular ellipse and a negative slope, indicating a more independent relationship between the bulk and surface densities. The larger uniformity of the snow samples and smaller spread of the density values in the melting stage, compared to the other stages, probably contributed to this weak relationship, as only melting, heavily metamorphosed snow crystals were observed

- Line 257: Clarify how you obtained 26 snow density points from a snowpack only 15 cm thick. What was the vertical resolution of measurements?
- The 26 snow density points in Figure S3 do not represent actual snow density measurements but instead show normalized snow depth. Since the snow depth at the snow pits varied a lot

during MOSAiC, we normalized the snow depth to facilitate better comparison of snow density profiles.

• Line 264–265: The observed drop in surface snow density from 300 to 150 kg/m³ in May is striking. What mechanism caused this transformation?

By looking at the density observations, we indeed see such a distinct decrease in snow density for both surface and bulk snow from April to early May. This characteristic was seen for the 10-day moving average and the optimal interpolation of daily observed snow density. The decrease of snow density in April-May may be explained by the fact that there is more abundant precipitation during spring months (Matrosov, et al., 2022).

The following text was given in the revised manuscript (L319-324):

Figure 8 shows the observed 10-day moving average and parameterised snow densities. In November, the variations in surface and bulk snow densities are opposite. From December to April, both surface and bulk snow densities increased. Overall, the correlation between the surface and bulk densities is 0.73, and the surface snow density has a larger standard deviation than the bulk snow density. In late April to early May, the surface snow density revealed a large drop from 320 kg/m3 to 190 kg/m3 with a standard deviation of 48 kg/m3. This change may be attributed to the abundant precipitation during spring months (Matrosov et al., 2022).

• Line 339 (Fig. 8): Why is there a drop in both bulk and surface snow densities in April and May? Also, if you initialized the E1–E3 simulations with 250 kg/m³, why do the first values on those curves begin at ~125 kg/m³?

In Table 1, the definitions of  $\rho_0$  and  $\rho_m$  as "initial snow density" and "maximum bulk density" were inaccurate and misleading. They rather represented "baseline snow density", e.g., fresh snow under minimal forcing, and the reference snow density response under the maximum environmental forcings, respectively. After the impact of environmental temperature and DOY, the snow density follows its characteristics. We have modified the text in Table 1 accordingly.

Line 345 (Table 2): Add the initial densities used for each E1–E3 simulation scheme.

The name "initial density" is not accurate (see previous response). It should be the "baseline snow density", and we added it in Table 2.

• Lines 415–417: Could you please add some more details here on how you calculated and compared this? It's not quite clear how 330 kg/m3 is related to the bulk values measured using the 10-day average. By snow accumulation do you mean snow depth?

Apologies for the ambiguity. This snow density (330 kg/m³) is unrelated to the observed 10-day average snow density. Yes, 'snow accumulation' here refers to snow depth.

Precipitation is an external forcing for the HIGHTSI snow and ice modelling. Precipitation is given as SWE in [mm/h] in snow water equivalent. We need to convert it into snow depth. For this reason, a bulk snow density of 330 kg/m3 was used to convert SWE to snow depth in [m]. This criterion was applied in previous studies (e.g., Huwald, et al., 2005, Cheng et al., 2008, Wang et al., 2015). For an existing snow layer, the bulk snow density ρb was determined by applying the observed 10-day moving average (black dashed line in Fig. 8).

We have modified the description to make it clearer:

A constant snow density of 330 kg/m³ was used to convert external forcing of precipitation to snow depth accumulation. For the existing snow, the bulk snow density ρb was the observed 10-day moving average (black dashed line in Fig. 8)

The following text was used in the revised manuscript (L165-168):

The snow precipitation (expressed in SWE) needs to be converted to snow depth (m) as model input. A density of 340 kg/m3 (denoted as  $\rho$ \_s0 in Table S1) was used for the conversion based on mass conservation. For the snow layer on sea ice, snow density is either derived from MOSAiC observations or parameterised.

• Line 441: Separate the **Discussion** and **Conclusion** sections. The current Conclusion is overly long and makes it hard to distill the key outcomes.

We separated the discussion and conclusion accordingly. The conclusion has been reformulated for better clarity and consistency.

Both chapters are largely reformulated and revised.

• Lines 526–527: This statement is debatable. Consider rephrasing or providing supporting evidence.

Higher densities over thin snow layers are due to the stronger wind compaction over smooth ice, while thicker snow is less dense because of the loose depth hoar bottom (Wagner et al., 2022)

The following text was used in the revised manuscript (L504-509)

During the winter-spring period, the mean spatial standard deviation of the surface snow density (77 kg/m3) was larger than that of the bulk snow density (59 kg/m3). This illustrates the strong spatial variability of the surface snow properties (Fig. 8). King et al., (2020) observed that snow density is highest in thin snow layers over undeformed ice and lowest in thicker snow layers over older and deformed ice. Higher densities over thin snow layers are due to the stronger wind compaction over smooth ice, while thicker snow is less dense because of the loose depth hoar bottom (Wagner et al., 2022). This depth hoar is caused by metamorphism occurring when the snowpack is exposed to temperature gradients (King et al., 2020).

**Final Remarks:**

Overall, this paper presents valuable insights into the evolution of snow density on Arctic sea ice and has the potential to make a solid contribution to the field. However, several sections require reorganization, clarification, and proofreading. Specific improvements in figures, methods, and interpretation would significantly enhance the quality and clarity of the manuscript.

Thank you for the constructive comments. We have carried out a major revision of the manuscript according to the comments from both reviewers. A summary of the major revisions is provided in the cover letter.

**Comments from Referee 2**

Based on sampling and measurement data from MOSAiC snow pits and thermodynamic models of sea ice, the authors investigated the seasonal variation of snow density and possible influencing factors, revealing the impact of snow density on the thermodynamic process of sea ice. The parameterization scheme of snow density and its simulation effect were evaluated, and the results can further support the optimization of the parameterization scheme of the sea ice component model in the Earth System model. It is a work worth sharing in the sea ice and even Arctic climate research communities. However, currently, the manuscript needs further improvement in terms of terminology, methodology, and analysis of results. Therefore, I recommend that the paper undergo major revisions before considering publication in TC.

**Major comments:**

1) The study evaluated four parameterization schemes for snow density, but did not provide recommendations on which scheme is the preferred one or what optimization is needed to better reproduce the observed seasonal changes in snow density.

One of the original objectives of this study was to identify and recommend an existing snow density parameterisation for snow/sea ice and climate modelling studies, and even to derive a new snow density parameterisation scheme based on MOSAIC data. However, we found that substantial spatiotemporal variability in snow density observations, along with physical snow sampling constraints and discontinuities in MOSAiC ice camp drift, prevented the derivation of a robust new snow density parameterization scheme. In addition, considering the MOSAiC campaign represents a specific ice drift trajectory, to have a representative snow density change along the Arctic sea ice transpolar drift corridor, one would perhaps need snow density data along several transpolar drift trajectories. Only in this way we would obtain an adequate full picture on snow density temporal spatial time series to derive a new snow density scheme.

Therefore, we focused on assessing the performance of the existing snow density schemes and pointed out their weaknesses. However, following our initial objective and the reviewer's comment, we provided a quantitative assessment of these snow density parameterisations (Tab. 2). Based on error statistics, we concluded that snow density parameterisations E3 and E4 perform better than the others. We have now added this statement in the Conclusions and outlook section. The snow density prognostic equation developed by Anderson (1976) remains applicable for sea ice thermodynamic modelling. We emphasised the need for further optimisation (e.g., addressing spatial variability) to improve snow density parameterisations. We have improved the Conclusions and outlook accordingly.

The following text was given in the revised manuscript (L545-546)

Based on error statistics, snow density parameterisations E3 and E4 performed better than the others and are recommended for modelling applications.

2) MOSAiC is an observation focused on the seasonal evolution of snow and sea ice physical processes. However, its seasonal evolution also overlaps with spatial-change information, especially after spring, as the ice floe drifts southward in the transpolar stream region, significant changes in atmospheric and oceanic conditions occur. Therefore, the representativeness of the observation results and parameterized evaluations for the pan Arctic Ocean needs further discussion. In addition, the MOSAiC floe in the study year has the characteristics of faster southward drift and lower snowfall compared with other years for the sea ice in the same location. How do these factors affect the representativeness of the observation results of snow density?

删除[Yubing Cheng]:

Thank you for this comment. The snow density time series along the MOSAiC drift trajectory includes both temporal (October-August) and spatial variability (large-scale ice drift and sampling sites distributed around the ice camp). This spatiotemporal variation occurred simultaneously throughout the campaign. The observed snow density evolution likely differs from seasonal snow density patterns at fixed locations, primarily due to contrasting weather regimes between the central Arctic and the marginal ice zone near the Fram Strait. We have added text in the Discussion section to highlight these differences.

We agree that the MOSAiC snow density observations are specific to a single drift trajectory and the environmental conditions of the studied ice floe. The accelerated southward drift and reduced snowfall during our study period significantly influenced snow metamorphic processes and density evolution. These conditions likely differ substantially from snow metamorphism patterns characteristic of the central Arctic.

On the other hand, we argue that sea ice in the central Arctic is not isolated, and the ice moves continuously along the transpolar drift corridor. The Lagrangian approach—observing the same ice floe over a period of time—allows us to isolate the temporal evolution of snow density under observed meteorological forcing, although the forcing is specific to the location of the ice floe. To fully distinguish between temporal and large-scale variations is not possible on the basis of data from a drifting ice station.

Although absolute snow density values exhibit regional variability, we maintain that the process-level relationships between snow density and meteorological drivers remain robust within the Transpolar Drift region. These relationships are well-constrained by consistent co-located observations of snow properties and meteorological variables across the ice floe. However, we recognize that pan-Arctic extrapolation requires caution, and recommend that future studies incorporate data from diverse drift regimes to enhance parameterization schemes.

We have added discussions on the representativeness of the snow density by comparing it with other observations.

In the beginning of the revised Discussion chapter 4, we have compared the MOSAiC snow density with other observations. The entire Chapter 4 was revised substantially.

The following text was given in the revised manuscript (L525-530)

MOSAiC snow density observations are constrained to a single drift trajectory and the specific environmental conditions of the studied ice floe. However, snow density along the MOSAiC drift trajectory captured both temporal (October-August) and spatial (distributed sampling sites) variability throughout the campaign. The Lagrangian approach—tracking the same ice floe over a period—allowed us to isolate the temporal evolution of snow density under observed meteorological forcing. However, we should not overemphasise the representativeness of the MOSAiC drift trajectory. To understand the overall snow density evolution along the Transpolar Drift Stream, more in-situ observations are needed.

**Special comments**

1) Line 20 "The modelled mean surface temperature" what is the surface temperature here? snow or sea ice surface?

This refers to the snow surface temperature

2) Figure 1: there are two sites that do not belong to MOSAiC CO, but the two temporary ice stations implemented after the vessel finished the drifting.

We have described it in the revised Figure caption.

The following text in the caption for figure 1 (L86-89)

Figure 1: (a) The MOSAiC ice camp drift trajectory during legs 1, 2, and 3. (b) Evolution of observed mean bulk snow density with respect to time and latitude. The colour bar illustrates the density values in both (a) and (b). Note that observations were also taken out of the ice camp trajectory at temporary stations (individual dots in both plots) established after the MOSAiC expedition ended on 17 September 2020.

3) Line 207 and other relative context: "According to the annual cycle of air temperature, we can categorize four periods": If we only look at the regime of air temperature, the second stage should last until early April. In fact, the near-surface air temperature remained at a relatively low level until early April. The snow density also began to increase only after early April.

There was debate among the co-authors regarding the definition of stages 1 and 2. The air temperature (Fig. 4b) exhibited a consistent decreasing trend from 27 October to 16 February, followed by a winter warm event (likely storm-associated) between 16 February and 3 March. From 3 March onward until 9 May, temperatures increased steadily. The key question was whether to assign the February/March winter warm event to Stage 1 or Stage 2. For clarity, we ultimately divided Stages 1 and 2 to align with MOSAiC legs 2 and 3. This separation did not impact our data analyses.

The following text was given in the revised manuscript (L203-208)

The cold season (defined as Stage I) starts from the beginning of the observations and lasts until 18 February. During this period, the mean air temperature was -25 °C, ranging between -7 °C and -38 °C. The warming period (Stage II) lasts from February 19 to May 10. Within this period, the air temperature increases from -40 °C to -10 °C. The temperature fluctuation between 16 February and 3 March (Fig. 4b) is associated with cyclone passages (Aue et al., 2023). For clarity, we ultimately divided Stages I and II to align with MOSAiC legs 2 and 3. This separation of Stages I and II did not impact our data analyses.

4) Line 230: "with the highest and lowest wind speeds recorded at 13.8 m/s and 1.3 m/s" Is this a daily value? If it's a real-time value, the highest of 13 m/s seems too small.

Yes, these are the daily average wind speed values, but not the real-time values. For the real hourly mean, the highest and lowest wind speeds were 16.3 m/s and 0.2 m/s, respectively. We clarified the meaning of the given values in the revised text.

The following text was given in the revised manuscript (L226)

The average wind speed during the study period was 5.9 m/s, with the highest and lowest daily mean values of 13.8 m/s and 1.3 m/s, respectively.

5) Lines 249-253: Can you introduce the situation of the third stage before discussing the situation of the fourth stage?

Yes, we revised the text order and discussed Stage III before Stage IV.

- 6) 256 "in Stage I, II, II and IV, respectively" This should be a typing mistake. Corrected.
- 7) The cumulative wind speed: When did this start to be calculated? How to distinguish the impacts on the fresh snow and on the old snow?

The calculation started from the beginning of the snow observation period.

The primary goal was to identify the effect of wind on both surface and bulk snow samples. The methodology did not differentiate wind effects on fresh snowfall and old snowpack. To distinguish the impact of wind on fresh and old snow, successive collections of samples of new and old snow over a long period would be necessary, which were unfortunately not available during the MOSAiC expedition.

8) What is the physical mechanism behind the decrease in snow density since April, and why can't all four parameterization schemes describe this mechanism? Can we optimize the parameterization scheme to reproduce this physical process?

This is a good question. By looking at the density observations, we indeed see such a distinct decrease in snow density for both surface and bulk snow since April. This characteristic was seen for the 10-day moving average and the optimal interpolation of snow density. The decrease of snow density in April may be explained by the fact that there is more abundant precipitation during spring months (Matrosov et al., 2022). We confirm that those snow density schemes did not take into account the impact of new snowfall. Optimizing new snowfall on density parameterization would require more in situ observations. We would rather leave this effort for future research. In the revised text, we added an explanation for the time series of observed densities.

The following text was given in the revised manuscript (L319-326)

Figure 8 shows the observed 10-day moving average and parameterised snow densities. In November, the variations in surface and bulk snow densities are opposite. From December to April, both surface and bulk snow densities increased. Overall, the correlation between the surface and bulk densities is 0.73, and the surface snow density has a larger standard deviation than the bulk snow density. In late April to early May, the surface snow density revealed a large drop from 320 kg/m³ to 190 kg/m³ with a standard deviation of 48 kg/m³. This change may be attributed to the abundant precipitation during spring months (Matrosov et al., 2022).

9) Figure. 9: Does it mean that the response of such thermodynamic parameters of sea ice to changes in constant snow density is close to linear?

Yes, the reviewer is correct. For a constant snow density, the response of modelled thermodynamic sea ice parameters to increasing snow density is linear.

- 10) Table 4: Can you provide specific details on the measurement principle of P1-P6? What is the underlying principle of their differences?
- P1 P6 were the total snow water equivalent (SWE) observed by different sensors for our simulation period. The measurements were made by various sensors, such as, vertically pointing 35-GHz Doppler radar, optical sensor, optical disdrometer, and weighing gauge. These instruments were placed at different locations around the MOSAiC central observatory (CO).

The measurement technologies and instrumentation calibration are responsible for explaining the differentiation of the observed results. For the physical principle, we believe the following processes are likely to contribute to the differences in observed SWE: 1) The local spatial variability of snow precipitation; 2) The wind effect on snow drift; 3) The wind effect on the blowing of snowfall.

We have merged the key message of the text above into the revised manuscript.

The technical details of SWE observations are:

Vaisala PWD22 optical sensors: See Kyrouac and Holdridge (2019); PARSIVEL-2 optical disdrometers: Nemeth and Beck (2011) and Wang et al. (2019); Pluvio weighing bucket precipitation gauge: Nemeth (2008) and Bartholomew (2020); Aerosol Observing System (AOS) precipitation sensor and Siphon gauge: Kyrouac and Springston (2019).

We have described the precipitation measurement in section 2.1 "Weather data". More information on the snow precipitation measurement was given in the Table 4 caption:

Table 4: Modelled mean snow and ice thermodynamic parameters using snow precipitation (SWE) accumulated during the modelling period. Model runs P1-P6 were carried out using the monthly snow accumulation data listed in Table 1 of Matrosov et al. (2022), which were obtained with different sensors: the vertically pointing 35-GHz Doppler radar (KAZR) with a range gate height of 0.17-km AGL onboard the RV Polarstern (Run P1), the Present Weather Detector (PWD) onboard the RV Polarstern (Run P2), the PWD at the CO ice camp (Run P3), the Pluvio precipitation gauge at the CO ice camp (Run P4), the Particle Size and Velocity (PARSIVEL-2) optical disdrometer onboard the RV Polarstern (Run P5), and the KAZR with a range gate height of 0.23-km AGL onboard the RV Polarstern (Run P6).

11) Line 438 "the above processes due to snow-to-ice transformation via surface flooding and resulting snow-ice formation": Did you consider the formation of snow ice in your thermodynamic simulation?

Yes, the snow-to-ice transformation was included in the model. For runs with different snow densities, snow ice did not form because the snow depth was relatively thin compared to the initial ice thickness. However, in precipitation sensitivity runs P3, P4, snow-ice formation occurred because precipitation led to significant snow accumulation.

The following text was given in the revised manuscript (L448-451)

However, a further increase in snow accumulation would reverse the above processes due to snow-to-ice transformation. For example, runs P3 and P4 yielded a net of 0.08 m and 0.15 m snow-ice, respectively, because the initial ice thickness was relatively thin (see Table S1) and the accumulated precipitation was large (see Table 4), the net increase in ice thickness (run P4) began to increase (Fig. 10d).

Thank you for the constructive comments. We have carried out a major revision of the manuscript according to the comments from both reviewers. A summary of the major revisions is provided in the cover letter.

**Reference:**

Anderson EA.: A point energy and mass balance model of a snow cover, Office of Hydrology, National Weather Service, Maryland, NOAA Technical Report NWS19, 1976.

Bartholomew, M.: Weighing bucket rain gauge instrument handbook. Available at https://www.arm.gov/publications/tech reports/handbooks/ doe-sc-arm-tr-232.pdf, 2020.

Bormann, K. J., Westra, S., Evans, J. P., and McCabe, M. F.: Spatial and temporal variability in seasonal snow density, Journal of Hydrology, 484, 63–73, https://doi.org/10.1016/j.jhydrol.2013.01.032, 2013.

Cheng, B., Zhang, Z., Vihma, T., Johansson, M., Bian, L., Li, Z., and Wu, H.: Model experiments on snow and ice thermodynamics in the Arctic Ocean with CHINARE 2003 data, Journal of Geophysical Research: Oceans, 113, https://doi.org/10.1029/2007JC004654, 2008.

Helfricht, K., Hartl, L., Koch, R., Marty, C., and Olefs, M.: Obtaining sub-daily new snow density from automated measurements in high mountain regions, Hydrology and Earth System Sciences, 22, 2655–2668, https://doi.org/10.5194/hess-22-2655-2018, 2018.

Huwald, H., Tremblay, L.-B., and Blatter, H.: Reconciling different observational data sets from Surface Heat Budget of the Arctic Ocean (SHEBA) for model validation purposes, Journal of Geophysical Research: Oceans, 110, https://doi.org/10.1029/2003JC002221, 2005.

Judson, A. and Doesken, N.: Density of Freshly Fallen Snow in the Central Rocky Mountains, Bulletin of the American Meteorological Society, 81, 1577–1588, https://doi.org/10.1175/1520-0477(2000)081<1577:DOFFSI>2.3.CO;2, 2000.

Kyrouac, J, Holdridge, D.: Surface meteorological instrumentation (PWD). ARM mobile facility (MOS). Available at https://www.arm.gov/publications/ tech\_reports/handbooks/met\_handbook.pdf. http://www.archive.arm.gov,2019.

Kyrouac, J, Springston, S.: Meteorological Measurements associated with the Aerosol Observing System (AOSMET). Atmospheric Radiation Measurement (ARM) user facility. DOI: http://dx.doi.org/10.5439/1025153, 2019.

Macfarlane, A. R., Schneebeli, M., Dadic, R., Tavri, A., Immerz, A., Polashenski, C., Krampe, D., Clemens-Sewall, D., Wagner, D. N., Perovich, D. K., Henna-Reetta, H., Raphael, I., Matero, I., Regnery, J., Smith, M. M., Nicolaus, M., Jaggi, M., Oggier, M., Webster, M. A., Lehning, M., Kolabutin, N., Itkin, P., Naderpour, R., Pirazzini, R., Hämmerle, S., Arndt, S., and Fons, S.: A Database of Snow on Sea Ice in the Central Arctic Collected during the MOSAiC expedition, Sci Data, 10, 398, https://doi.org/10.1038/s41597-023-02273-1, 2023.

Matrosov, S. Y., Shupe, M. D., and Uttal, T.: High temporal resolution estimates of Arctic snowfall rates emphasizing gauge and radar-based retrievals from the MOSAiC expedition, Elementa: Science of the Anthropocene, 10, 00101, https://doi.org/10.1525/elementa.2021.00101, 2022.

Nemeth, K, Beck, E.: Precipitation measurement. Meteorological Technology International Magazine 2011: 105–107, 2011.

Nemeth, K.: OTT Pluvio2: Weighing Precipitation Gauge and Advances in Precipitation Measurement Technology. BDM Meteorology OTT MESSTECHNIK GmbH & Co. KG Ludwigstr. 16, 87437. Kempten, Germany. Available at https://www.wmo.int/pages/prog/www/IMOP/publications/IOM-96\_TECO-2008/P2(18)\_Nemeth\_Germany.pdf, 2008.

Rennert, K. J., Roe, G., Putkonen, J., and Bitz, C. M.: Soil thermal and ecological impacts of rain on snow events in the circumpolar arctic, Journal of Climate, 22, 2302–2315, https://doi.org/10.1175/2008JCLI2117.1, 2009.

Smith, M. M., Light, B., Macfarlane, A. R., Perovich, D. K., Holland, M. M., and Shupe, M. D.: Sensitivity of the Arctic Sea Ice Cover to the Summer Surface Scattering Layer, Geophysical Research Letters, 49, e2022GL098349, https://doi.org/10.1029/2022GL098349, 2022.

Stroeve, J., Nandan, V., Willatt, R., Dadic, R., Rostosky, P., Gallagher, M., Mallett, R., Barrett, A., Hendricks, S., Tonboe, R., McCrystall, M., Serreze, M., Thielke, L., Spreen, G., Newman, T., Yackel, J., Ricker, R., Tsamados, M., Macfarlane, A., Hannula, H.-R., and Schneebeli, M.: Rain on snow (ROS) understudied in sea ice remote sensing: a multi-sensor analysis of ROS during MOSAiC (Multidisciplinary drifting Observatory for the Study of Arctic Climate), The Cryosphere, 16, 4223–4250, https://doi.org/10.5194/tc-16-4223-2022, 2022.

Sturm, M., Perovich, D. K., and Holmgren, J.: Thermal conductivity and heat transfer through the snow on the ice of the Beaufort Sea, Journal of Geophysical Research: Oceans, 107, SHE 19-1-SHE 19-17, https://doi.org/10.1029/2000JC000409, 2002a.

Sturm, M., J. Holmgren, and D. K. Perovich, Winter snow cover on the sea ice of the Arctic Ocean at the Surface Heat Budget of the Arctic Ocean (SHEBA): Temporal evolution and spatial variability, J. Geophys. Res., 107(C10), 8047, doi:10.1029/2000JC000400, 2002b.

Svensson, G., Murto, S., Shupe, M. D., Pithan, F., Magnusson, L., Day, J. J., Doyle, J. D., Renfrew, I. A., Spengler, T. and Vihma, T.: Warm air intrusions reaching the MOSAiC expedition in April 2020—The YOPP targeted observing period (TOP), Elementa: Science of the Anthropocene, 11 (1). doi: 10.1525/elementa.2023.00016, 2023.

Wagner, D. N., Shupe, M. D., Cox, C., Persson, O. G., Uttal, T., Frey, M. M., Kirchgaessner, A., Schneebeli, M., Jaggi, M., Macfarlane, A. R., Itkin, P., Arndt, S., Hendricks, S., Krampe, D., Nicolaus, M., Ricker, R., Regnery, J., Kolabutin, N., Shimanshuck, E., Oggier, M., Raphael, I., Stroeve, J., and Lehning, M.: Snowfall and snow accumulation during the MOSAiC winter and spring seasons, The Cryosphere, 16, 2373–2402, https://doi.org/10.5194/tc-16-2373-2022, 2022.

Wang, D, Bartholomew, M, Shi, Y.: Atmospheric radiation measurement (ARM) user facility. ARM Mobile Facility (MOS) MOSAiC. Laser Disdrometer (LD). DOI: http://dx.doi.org/10.5439/1779709, 2019.

Wang, T., Peng, S., Ottlé, C., and Ciais, P.: Spring snow cover deficit controlled by intraseasonal variability of the surface energy fluxes, Environ. Res. Lett., 10, 024018, https://doi.org/10.1088/1748-9326/10/2/024018, 2015.

---

## Referee Report (RR1)

**Line 15** A numerical snow and sea ice model was applied to simulate the sensitivity of sea ice to snow density and snow precipitation during the period when snow was dry.

I may be mistaken, but I could not find any other reference to "dry snow" in the manuscript. Please clarify this in the modelling section, as the condition under which the simulations were conducted is not entirely clear.

**Lines 18–19** The examined snow density schemes produced mean snow densities consistent with MOSAiC observations; however, none of the schemes adequately captured the observed temporal variability in snow density.

From the manuscript, it is clear that the densification schemes you selected did not reproduce the temporal evolution of the snowpack. However, snow compaction schemes implemented in models such as SNOWPACK and CROCUS have demonstrated good agreement with in-situ data. Could you clarify why you did not apply those schemes in your study?

**Lines 116–119, 115–140** During MOSAiC, comprehensive sea ice and snow observations were carried out (Nicolaus et al., 2022). Snow pit measurements were taken at least weekly but often on several days per week, and occasionally more than once a day. Snow pits were dug at various locations on undeformed first-year ice, second-year ice, and places close to open leads and pressure ridges. Most measurements were taken within the central observatories in designated clean, undisturbed snowfields.

In this section you describe the frequency and distribution of snow pit measurements. However, it remains unclear how many pits were dug at each type of location (undeformed first-year ice, second-year ice, ridges, leads, etc.). This information is crucial, as snow properties differ substantially between these environments. For example, repeated measurements near ridges could bias the calculated mean depth and density relative to undeformed first-year ice. I recommend specifying the number of pits at each location and, ideally, providing an estimate of the relative areal contribution of these representative sites to the total study region. This would help readers assess how representative the aggregated values truly are.

---

## Author Response (AR2)

Dear Editor Masashi Niwano

Thank you for handling our manuscript. We have made further improvements to the manuscript based on comments from you and reviewers

Please see our response (text in blue) below

Please let us know if further improvements are still needed.

Best regards,

Yubing Cheng, Bin Cheng and co-authors
* * *
Dear Yubing Cheng, Bin Cheng, and coauthors,

Thank you for submitting the revised manuscript together with the detailed response letter. Anonymous Referee #1 reviewed the revised manuscript, and the referee is generally satisfied with your responses, although the referee has provided some additional minor comments to improve the paper further. Regarding your responses to Anonymous Referee #2, I have checked them along with the relevant revisions and judged that they are convincing. Based on the results, I have judged that this paper can be published after minor revisions. Please consider the constructive comments provided by Anonymous Referee #1 and revise the paper accordingly. Below, I also list some technical comments, which I would like to ask you to consider:

- Figure 3: It seems to me that it is not referred to in the running text. Please check.

The text on lines 97-99 was used to explain Figure 3. The location and the style of expression may be a bit confusing. So, we modified the text:

The relationship between SWE (mm) and snow depth (m) is illustrated in Figure 3. The regression slopes represent mean snow densities for each period: 348 kg/m³ (entire MOSAiC campaign), 308 kg/m³ (winter-spring period), and 487 kg/m³ (MOSAiC legs 4 and 5). These values likely reflect the characteristic snow or surface scattering layer densities at a specified period.

- Tables S1 and S2: It is better to change the order of these tables, because Table S2 is referenced in the running text earlier than Table S1 (L. 105, manuscript v3). If you agree with this point, please revise the following places accordingly: L. 105, L. 166, L. 168, and L. 447.

**Agreed, we made changes accordingly.**

- Figure 7a: Because the line thicknesses/colors for the linear regression coefficient curves and the error bars are almost the same, it is a bit difficult to distinguish them. Can you improve the presentation of this figure?
- Figure 7b: It is better to add an explanation for the linear regression lines in the caption.

We redraw Figure 7a using the y-axis from both sides of the figure. The regression lines are shown separately with better visualisation.

The figure 7 caption for (b) is updated:

(b) Example for 3 days wind speed accumulation versus surface(red) and bulk(black) snow densities. The solid lines represent the ordinary least squares linear regression fits.

The manuscript addresses an important topic, but several points require clarification before acceptance. First, the use of the term "dry snow" (Line 15) is unclear, as it is not consistently referenced elsewhere; the modelling section should explicitly state the conditions under which the simulations were conducted. Second, while the evaluated snow density schemes reproduce mean values reasonably well, they fail to capture temporal variability. Given that models such as SNOWPACK and CROCUS include more advanced compaction schemes with demonstrated agreement with in-situ data, the authors could clarify why these were not applied. Finally, in the description of MOSAiC snow pit observations (Lines 115–140), the manuscript lacks detail on the distribution of pits across different ice environments. Since snow properties vary substantially between undeformed ice, ridges, and leads, this information is crucial to assess representativeness. Providing the number of pits at each location and their relative areal coverage would strengthen the credibility of the observational dataset.

Line 15 A numerical snow and sea ice model was applied to simulate the sensitivity of sea ice to snow density and snow precipitation during the period when the snow was dry. I may be mistaken, but I could not find any other reference to "dry snow" in the manuscript. Please clarify this in the modelling section, as the condition under which the simulations were conducted is not entirely clear.

**We added text in 2.4 Snow and ice model to clarify this matter:**

A single-column high-resolution thermodynamic snow and ice model (HIGHTSI) is used to simulate the sensitivity of snow density on the thermal regime and mass balance of snow and ice during the winter-spring period (28 October 2019 - 6 May 2020) when the air temperature was below zero degrees and snow was dry.

Lines 18–19 The examined snow density schemes produced mean snow densities consistent with MOSAiC observations; however, none of the schemes adequately captured the observed temporal variability in snow density.

From the manuscript, it is clear that the densification schemes you selected did not reproduce the temporal evolution of the snowpack. However, snow compaction schemes implemented in models such as SNOWPACK and CROCUS have demonstrated good agreement with in-situ data. Could you clarify why you did not apply those schemes in your study?

This is a good comment, and thank you for pointing out this issue.

The reasons we did not apply those advanced snow schemes are: 1) we are aware of SNOWPACK and CROCUS models. As far as we understand, those are advanced snow process models that have been applied to alpine snow or terrestrial snow where the snow densities differ from those in the polar regions. The parameters applied in the snow compaction schemes may not be valid for the polar conditions; 2) We focus on assessing snow density schemes that are computationally cheap, that can be applied for climate research. So, the snow density schemes we picked up are simple, and they use adequate input snow parameters that are available. On the other hand, we acknowledge that advanced snow models could be valuable for climate research once their computational efficiency is optimised and snow observations are improved.

Lines 116–119, 115–140 During MOSAiC, comprehensive sea ice and snow observations were carried out (Nicolaus et al., 2022). Snow pit measurements were taken at least weekly but often on several days per week, and occasionally more than once a day. Snow pits were dug at various locations on undeformed first-year ice, second-year ice, and places close to open leads and pressure ridges. Most measurements were taken within the central observatories in designated clean, undisturbed snowfields. In this section you describe the frequency and distribution of snow pit measurements.

However, it remains unclear how many pits were dug at each type of location (undeformed first-year ice, second-year ice, ridges, leads, etc.). This information is crucial, as snow properties differ substantially between these environments. For example, repeated measurements near ridges could bias the calculated mean depth and density relative to undeformed first-year ice. I recommend specifying the number of pits at each location and, ideally, providing an estimate of the relative areal contribution of these representative sites to the total study region. This would help readers assess how representative the aggregated values truly are.

We added sample numbers and their percentage to the total snow samples in the revised manuscript:

Snow pit measurements were taken at least weekly but often on several days per week, and occasionally more than once a day. Most measurements were taken within the central observatories in designated clean, undisturbed snow fields. Snow pits were dug at various locations on undeformed first-year ice (107, 142), multi-year ice (20, 76), and near open leads (16, 17), ponds (5, 2), and pressure ridges (61, 35). The numbers in parentheses indicate the sample counts for surface and bulk snow, respectively. More than half of the surface and bulk snow samples (51% and 52%) were collected over the first-year ice (FYI).